# 🦅 Falcon: Fast Visuomotor Policies via Partial Denoising

**Haojun Chen** [* 1 2]  **Minghao Liu** [* 3]  **Chengdong Ma** [1]  **Xiaojian Ma** [2]  **Zailin Ma** [4]  **Huimin Wu** [2]  **Yuanpei Chen** [1 5]
**Yifan Zhong** [1 5]  **Mingzhi Wang** [1]  **Qing Li** [2]  **Yaodong Yang** [1]

## Abstract

Diffusion policies are widely adopted in complex visuomotor tasks for their ability to capture multimodal action distributions. However, the multiple sampling steps required for action generation significantly harm real-time inference efficiency, which limits their applicability in real-time decision-making scenarios. Existing acceleration techniques either require retraining or degrade performance under low sampling steps. Here we propose Falcon, which mitigates this speed-performance trade-off and achieves further acceleration. The core insight is that visuomotor tasks exhibit sequential dependencies between actions. Falcon leverages this by reusing partially denoised actions from historical information rather than sampling from Gaussian noise at each step. By integrating current observations, Falcon reduces sampling steps while preserving performance. Importantly, Falcon is a training-free algorithm that can be applied as a plug-in to further improve decision efficiency on top of existing acceleration techniques. We validated Falcon in 48 simulated environments and 2 real-world robot experiments. demonstrating a 2-7x speedup with negligible performance degradation, offering a promising direction for efficient visuomotor policy design. The code is available at https://github.com/chjchjchjchjchj/Falcon.

## 1. Introduction

Diffusion policies have demonstrated remarkable success in addressing complex visuomotor tasks in robotics (Chi et al., 2023; Reuss et al., 2023; Ze et al., 2024b;a; Ravan et al., 2024; Yang et al.), thanks to their ability to model complex multimodal distributions and maintain stable training dynamics. Essentially, diffusion policies rely on the reverse sampling process of a stochastic differential equation (SDE) (Song et al., b; Karras et al., 2022), where actions are iteratively sampled starting from a standard normal distribution. Each sampling step involves drawing a sample from a Brownian motion distribution, incrementally denoising the initial sample to generate the final action. However, the iterative sampling process required for action generation makes these methods computationally slow, particularly when applied to sequential decision-making in real-time environments (Chen et al., b; Janner et al., 2022; Wang et al., b; Yang et al., 2023; Hansen-Estruch et al., 2023; Chen et al., a).

To address these challenges, existing work (Song et al., a;b; Lu et al., 2022; Zhang & Chen) reformulates the sampling process as an ordinary differential equation (ODE) and uses numerical solvers to reduce the number of denoising steps (Song et al., a; Lu et al., 2022; Zhang & Chen). However, these solvers often suffer from approximation errors when using very few steps, degrading policy performance (Zhao et al., 2024; Wang et al., a). Alternatively, distillation-based approaches (Salimans & Ho; Song et al., 2023; Kim et al.; Prasad et al., 2024; Wang et al., 2024) accelerate inference via one-step generation. However, they typically require task-specific retraining and may degrade multimodal expressiveness (Prasad et al., 2024), making them less flexible and unsuitable as plug-and-play solutions for diverse tasks. Another related approach is Streaming Diffusion Policy (SDP) (Høeg et al., 2024), which also uses partial denoising for acceleration but requires task-specific retraining and a large noise-time buffer, limiting flexibility and memory efficiency.

In this work, we introduce Falcon (**Fa**st visuomotor p**o**li**c**ies via partial den**o**isi**n**g), a novel approach designed to bridge the gap between acceleration and performance preservation in diffusion-based visuomotor policies. The key insight behind Falcon is that sequential dependencies in visuomotor tasks can be exploited to accelerate action generation while maintaining multimodal expressiveness. To effectively leverage this property, we first utilize the previous action prediction as the reference action, then introduce a thresholding

---

[*]Equal contribution  [1]Institute for Artificial Intelligence, Peking University  [2]National Key Laboratory of General Artificial Intelligence, BIGAI  [3]School of Electronic Engineering and Computer Science, Peking University  [4]School of Mathematical Sciences, Peking University  [5]PKU-PsiBot Joint Lab. Correspondence to: Qing Li <dylan.liqing@gmail.com>, Yaodong Yang <yaodong.yang@pku.edu.cn>.

*Proceedings of the 42nd International Conference on Machine Learning*, Vancouver, Canada. PMLR 267, 2025. Copyright 2025 by the author(s).

mechanism combined with one-step estimation to evaluate which partial denoising actions are dependent on current timesteps in parallel. Using a temperature-scaled softmax, we then select the most suitable partial denoising action to continue the sampling process, preserving both performance and efficiency. By avoiding the conventional practice of starting the denoising process from a standard normal distribution at each decision step, Falcon begins from partial denoised actions derived from historical observations, significantly reducing the number of sampling steps required. Importantly, Falcon is a training-free method, which allows it to be applied as a plug-in module, enhancing the efficiency of existing diffusion-based policies without additional training or extensive modifications. It integrates seamlessly with solvers like DDIM (Song et al., a) and DPMSolver (Lu et al., 2022) to further enhance acceleration.

We evaluate Falcon in both simulated and real-world environments. In simulation, we test Falcon across 48 tasks spanning five widely used benchmarks, including RoboMimic (Mandlekar et al., 2022), RoboSuite Kitchen (Gupta et al., 2020), BlockPush (Shafiullah et al., 2022), MetaWorld (Yu et al., 2020) and Maniskill2 (Gu et al., 2023). For real-world experiments, we deploy Falcon on a dual-arm robot platform in two manipulation tasks—dexterous grasping and high-precision insertion. In both settings, Falcon achieves a **2–7×** acceleration in action sampling while preserving high task success rates, validating its effectiveness and practical applicability.

In summary, this work makes three main contributions: **First,** we introduce Falcon, which leverages sequential dependencies in visuomotor tasks to perform partial denoising, significantly reducing the number of sampling steps while preserving the ability to model multimodal action distributions. **Second,** Falcon functions as a training-free plug-in that enhances existing diffusion-based policies, integrating seamlessly with solvers like DDIM and DPMSolver to further accelerate action generation. **Third,** we demonstrate Falcon's effectiveness across 48 simulated environments and 2 real-world robot tasks, achieving a **2-7×** speedup in inference while maintaining high action quality.

## 2. Preliminaries

In this section, we briefly introduce the diffusion models, how the diffusion models are used for diffusion policies in visuomotor tasks, the acceleration techniques in the diffusion-based models, and terminology.

**Diffusion Models.** Diffusion models, such as DDPM (Ho et al., 2020), are generative models that learn data distributions by progressively corrupting data through noise in a forward process and then reconstructing it in a reverse denoising process. In the forward process, a data point $x_0$

is corrupted K timesteps, resulting in:

$$q(\boldsymbol{x}_k \mid \boldsymbol{x}_0) = \mathcal{N}\left(\boldsymbol{x}_k; \sqrt{\bar{\alpha}_t}\boldsymbol{x}_0, (1 - \bar{\alpha}_k)\mathbf{I}\right), \quad (1)$$

where $\bar{\alpha}_t = \prod_{i=1}^{t} \alpha_i$. After $K$ steps, $\boldsymbol{x}_K$ becomes nearly Gaussian. The reverse process reconstructs $\boldsymbol{x}_0$ by iteratively denoising, modeled as

$$p_\theta(\boldsymbol{x}_{k-1} \mid \boldsymbol{x}_k) = \mathcal{N}\left(\boldsymbol{x}_{k-1}; \mu_\theta(\boldsymbol{x}_k, k), \sigma_t^2 \mathbf{I}\right). \quad (2)$$

where $\mu_\theta$ is the predicted mean, $\sigma_t^2$ is fixed according to the forward process, $\alpha_i$ is the variance schedule in the forward diffusion process.

**Diffusion Policies.** Diffusion policy (Chi et al., 2023) extends diffusion models as a powerful policy for visuomotor tasks. At timestep $t$, diffusion policy takes the latest $T_o$ steps of observation $\mathbf{O}_t$ as input, predicts $T_p$ action sequence $\mathbf{A}_t$ and executes $T_a$ action sequence. The action sequence generation process is a conditional denoising diffusion process modeled by $p_\theta(\mathbf{A}_t | \mathbf{O}_t)$, where $\mathbf{A}_t \in \mathbb{R}^{T_p \times D_a}$ and $\mathbf{O}_t \in \mathbb{R}^{T_o \times D_o}$, $D_a$ and $D_o$ represent the action and observation's dimension respectively. Specifically, starting from a pure Gaussian noise sample, diffusion policy leverage the noise prediction network $\varepsilon_\theta$ to predict and remove noise at each denoising step, iterating for $K$ steps to generate a clean sample $\mathbf{A}_t^0$. The action sequence generation process is described as Eq. 3, where $\mathbf{A}_t^k$ is a partial denoising action in the timestep $t$ with noise level $k$, $\mathbf{Z}$ is a standard Gaussian random variable.

$$\mathbf{A}_t^{k-1} = \frac{1}{\sqrt{\alpha_k}}\left(\mathbf{A}_t^k - \frac{1 - \alpha_k}{\sqrt{1 - \bar{\alpha}_k}}\boldsymbol{\epsilon}_\theta\left(\mathbf{O}_t, \mathbf{A}_t^k, k\right)\right) + \sigma_k \mathbf{Z}. \quad (3)$$

The training loss is

$$L(\theta) = \mathbb{E}_{k, \mathbf{A}_t^0, \boldsymbol{\epsilon}} \left[ \left\| \boldsymbol{\epsilon} - \boldsymbol{\epsilon}_\theta\left(\mathbf{O}_t, \mathbf{A}_t^k, k\right) \right\|^2 \right]$$
$$\mathbf{A}_t^k = \sqrt{\bar{\alpha}_k}\,\mathbf{A}_t^0 + \sqrt{1 - \bar{\alpha}_k}\,\boldsymbol{\epsilon} \quad (4)$$

**Acceleration techniques in the diffusion-based models.** Some popular works (Song et al., b; Karras et al., 2022; Song et al., a; Lu et al., 2022) interpret the diffusion model as an ODE. Specifically, DDIM (Song et al., a) generalizes the Markov forward process in DDPM to a non-Markov process, allowing the use of shorter Markov chains during sampling. The iterative steps of its sampling process can be rewritten in a form similar to Euler integration, which is a discrete solution process for a specific ODE. DPMSolver (Lu et al., 2022) accelerates sampling by analytically computing the linear part of the diffusion ODE solution and using an exponential integrator to approximate the nonlinear part. But these methods can reduce the quality degradation of few-step sampling (Shih et al., 2024). SDP (Høeg et al., 2024) improves the sampling speed by generating partial denoised actions with different noise levels, but relies on

handcrafted noise schedules and large buffers, limiting its flexibility. Consistency Policy (CP) (Prasad et al., 2024) distills a pretrained diffusion policy into a one-step sampler via self-consistency on its probability flow ODE. While fast, this requires retraining and may compromise multimodality.

We write $[a, b)$ to denote the set $\{a, a+1, \cdots, b-1\}$, $\boldsymbol{x}_{a:b}$ to denote the set $\{\boldsymbol{x}_i : I \in [a, b)\}$ and $[K]$ to denote the set $\{1, \cdots, K\}$. We define $\mathbf{A}_t = [\boldsymbol{a}_{t:t+T_a}, \tilde{\boldsymbol{a}}_{t+T_a:t+T_p}]$, where $\boldsymbol{a}_{t:t+T_a}$ is the executed part and $\tilde{\boldsymbol{a}}_{t+T_a:t+T_p}$ is the unexecuted part in timestep $t$.

# 3. Falcon: Fast Visuomotor Policies via Partial Denoising

In this section, we will introduce the core principles of Falcon. **First,** we present the way of leveraging previously generated action sequences $\tilde{\boldsymbol{a}}_{t:t-T_a+T_p}$, based on the model's confidence in its prior predictions. **Second,** we describe the thresholding mechanism that determines which partial denoised action from past timesteps serves as the initialization for the current denoising process. **Lastly,** we outline the implementation details of Falcon.

## 3.1. Reference Actions

Falcon first leverages the unexecuted action sequence $\tilde{\boldsymbol{a}}_{t:t-T_a+T_p}$, predicted from the previous observation $\mathbf{O}_{t-T_a}$, as a reference for denoising the current action sequence $\boldsymbol{a}_{t:t+T_p}$ at time step $t$. This approach is motivated by our observation that the Euclidean distance $\|\tilde{\boldsymbol{a}}_{t:t-T_a+T_p} - \boldsymbol{a}_{t:t-T_a+T_p}\|_2$ between $\tilde{\boldsymbol{a}}_{t:t-T_a+T_p}$ and $\boldsymbol{a}_{t:t-T_a+T_p}$ exhibits a high probability density near zero (see Fig. 2), indicating that $\tilde{\boldsymbol{a}}_{t:t-T_a+T_p}$ closely approximates $\boldsymbol{a}_{t:t-T_a+T_p}$ in most cases.

Furthermore, as shown in Eq. 4, diffusion policies take the latest $T_o$ observations as input and train to output the future $T_p$ action sequence $\mathbf{A}_t = [\boldsymbol{a}_{t:t+T_a}, \tilde{\boldsymbol{a}}_{t+T_a:t+T_a+Tp}]$. This training paradigm provides confidence that the predicted action sequence $\tilde{\boldsymbol{a}}_{t:t-T_a+T_p} \in \mathbf{A}_t$ is sufficiently accurate for use as a reference. Additionally, Falcon incorporates the score vector field $\nabla_{\mathbf{A}_t} \log p(\mathbf{A}_t|\mathbf{O}_t)$, computed from the current observation $\mathbf{O}_t$, to guide the denoising process of the partial denoised action at the current time step, ensuring more precise and stable action refinement.

## 3.2. Thresholding Mechanism

To determine which partial denoised action from historical observations should be used as the initialization for denoising at the current timestep, we employ Tweedie's approach (Efron, 2011; Chung et al.; Kim & Ye, 2021). Given that the forward process of diffusion policy follows

$$\mathbf{A}_t^k = \sqrt{\bar{\alpha}_k}\mathbf{A}_t^0 + \sqrt{1-\bar{\alpha}_k}\boldsymbol{z}, \ \ \boldsymbol{z} \sim \mathcal{N}(\mathbf{0}, \mathbf{I}), \quad (5)$$

We derive the posterior expectation, which serves as the one-step estimation of the partial denoised actions $\boldsymbol{a}_{\tau:\tau-T_a+T_p}^{k_i}$, denoted by $\hat{\boldsymbol{a}}_{\tau:\tau-T_a+T_p}^{k_i}$:

$$\hat{\boldsymbol{a}}_{\tau:\tau-T_a+T_p}^{k_i} = \mathbb{E}[\boldsymbol{a}_{\tau:\tau-T_a+T_p}^0 \mid \mathbf{O}_t, \boldsymbol{a}_{\tau:\tau-T_a+T_p}^k]. \quad (6)$$

This estimation is conditioned on the current observation $\mathbf{O}_t$, as formulated in Proposition 3.1. Here, $\tau < t$ represents a historical decision step and $k$ denotes the noise level of the partial denoised action.

The dependency between past and current actions is measured by the Euclidean $\|\hat{\boldsymbol{a}}_{\tau:\tau-T_a+T_p}^{k_i} - \tilde{\boldsymbol{a}}_{t:t-T_a+T_p}\|_2$ between $\hat{\boldsymbol{a}}_{\tau:\tau-T_a+T_p}^{k_i}$ and the reference action $\tilde{\boldsymbol{a}}_{t:t-T_a+T_p}$. If this distance falls below a predefined threshold $\epsilon$, the partial denoised action $\boldsymbol{a}_{\tau:\tau+T_p}^k$ is selected, as it is likely to converge to the desired action $\boldsymbol{a}_{t:t+T_p}$.

**Proposition 3.1.** *(Tweedie's formula for denoising) Let $\boldsymbol{x}$ be a random variable with a probability distribution $p(\boldsymbol{x})$, and Let $\boldsymbol{x}_\sigma := \boldsymbol{x} + \sigma\boldsymbol{z}$, where $\boldsymbol{z} \sim \mathcal{N}(0, I_D)$ and $\sigma > 0$ is a known scalar. Then, the best estimate $\hat{\boldsymbol{x}}_\sigma$ of $\boldsymbol{x}$ in mean squared error, given the noisy observation $\boldsymbol{x}_\sigma$, is given by the formula:*

$$\hat{\boldsymbol{x}}_\sigma := \mathbb{E}_{p(\boldsymbol{x}|\boldsymbol{x}_\sigma)}[\boldsymbol{x}] = \mathbb{E}[\boldsymbol{x} \mid \boldsymbol{x}_\sigma] = \boldsymbol{x}_\sigma + \sigma^2\nabla_{\boldsymbol{x}_\sigma}\log p(\boldsymbol{x}_\sigma) \quad (7)$$

*Remark* 3.2. In the context of DDPM (Ho et al., 2020) whose diffusion forward process is $\boldsymbol{x}_t \sim \mathcal{N}(\sqrt{\bar{\alpha}_t}\boldsymbol{x}_0, (1-\bar{\alpha}_t)I_D)$, Tweedie's formula can be rewritten as

$$\mathbb{E}[\boldsymbol{x}_0 \mid \boldsymbol{x}_t] = (\boldsymbol{x}_t + (1-\bar{\alpha}_t)\nabla_{\boldsymbol{x}_t}\log p(\boldsymbol{x}_t))/\sqrt{\bar{\alpha}_t}. \quad (8)$$

Since $\nabla_{\mathbf{A}_t^k}\log p(\mathbf{A}_t^k \mid \mathbf{O}_t) = -\frac{\varepsilon}{\sqrt{1-\bar{\alpha}_k}} \approx \frac{\varepsilon_\theta(\mathbf{O}_t, \mathbf{A}_t^k, k)}{\sqrt{1-\bar{\alpha}_k}}$ (Ho et al., 2020; Chi et al., 2023), we can obtain the posterior expectation by

$$\mathbb{E}[\boldsymbol{a}_{\tau:\tau-T_a+T_p}^0 \mid \mathbf{O}_t, \boldsymbol{a}_{\tau:\tau-T_a+T_p}^k]$$
$$= \frac{\boldsymbol{a}_{\tau:\tau-T_a+T_p}^k - \sqrt{1-\bar{\alpha}_k}\varepsilon_\theta(\mathbf{O}_t, \boldsymbol{a}_{\tau:\tau-T_a+T_p}^k, k)}{\sqrt{\bar{\alpha}_k}} \quad (9)$$

Finally, we define the set of candidate actions as $\mathcal{S} = \{\boldsymbol{a}_{\tau:\tau+T_p}^k : \|\hat{\boldsymbol{a}}_{\tau:\tau-T_a+T_p}^k - \tilde{\boldsymbol{a}}_{\tau:\tau-T_a+T_p}\|_2 < \epsilon, \forall \tau < t, k \in [K]\}$ and sample the starting point through the following distribution

$$\mathbb{P}\left(\mathbf{A}_\tau^{k_s} = \boldsymbol{a}_{\tau:\tau+T_p}^{k_i}\right) = \frac{\exp\left(-k_i\mathbf{1}_{\mathcal{S}}\left(\boldsymbol{a}_{\tau:\tau+T_p}^{k_i}\right)/\kappa\right)}{\displaystyle\sum_{\substack{\tau < t \\ k \in [K]}}\exp\left(-k_j\mathbf{1}_{\mathcal{S}}\left(\boldsymbol{a}_{\tau:\tau+T_p}^{k_j}\right)/\kappa\right)},$$
$$(10)$$

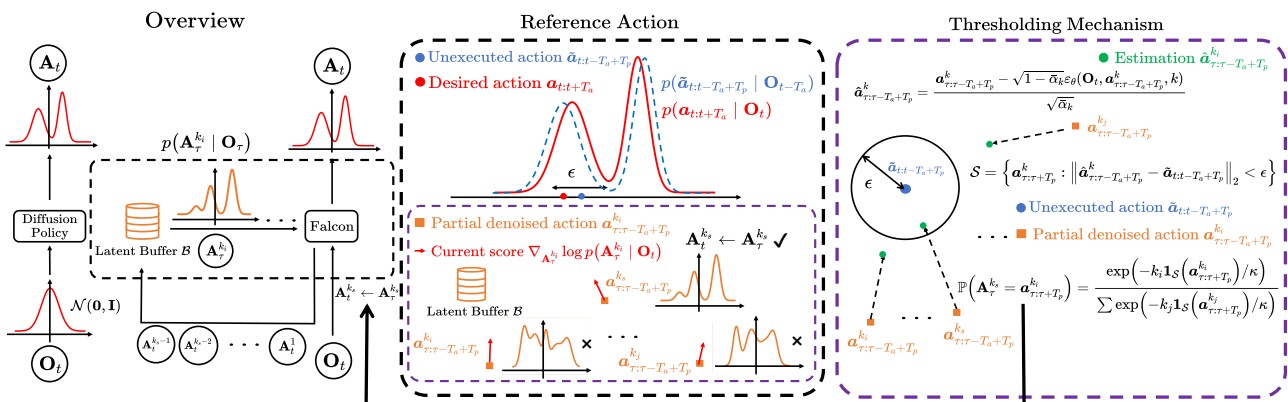

*Figure 1.* **Method Description** a) Falcon begins denoising from historically generated partial denoised actions rather than a normal distribution, requiring less than $k_s$ steps to produce the action sequence $\mathbf{A}_t$. The process involves 2 steps: setting reference actions and retrieving a partial denoised action sequence $\mathbf{A}_\tau^k$ from the latent buffer $\mathcal{B}$ to start denoising through a thresholding mechanism. b) Reference Action. Falcon uses unexecuted actions $\tilde{\boldsymbol{a}}_{t:t-T_a+T_p}$ from the previous step $t - T_a$ as the desired action $\boldsymbol{a}_{t:t+T_a}$, selecting a partial denoised action from $\mathcal{B}$ as the starting point. c) Thresholding Mechanism. Falcon evaluates all actions in $\mathcal{B}$ in parallel, identifying those close to the reference action after one-step estimation, and samples the starting point $\mathbf{A}_t^{k_s}$ based on the noise level.

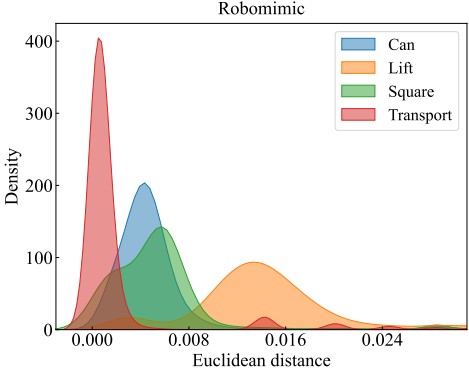

*Figure 2.* **Probability density estimation of** $\|\tilde{\boldsymbol{a}}_{t:t-T_a+T_p} - \boldsymbol{a}_{t:t-T_a+T_p}\|_2$. $\tilde{\boldsymbol{a}}_{t:t-T_a+T_p}$ is nearly the same as $\boldsymbol{a}_{t:t-T_a+T_p}$ since the majority of Euclidean distances between $\tilde{\boldsymbol{a}}_{t:t-T_a+T_p}$ and $\boldsymbol{a}_{t:t-T_a+T_p}$ are concentrated within the range of less than 0.015. Samples are collected by diffusion policy across four Robomimic environments (Can, Lift, Square, and Transport), where 200 trajectories were generated for each environment.

where $\kappa$ is the temperature scaling factor and $\mathbf{1}_{\mathcal{S}}(\cdot)$ is the indicator function of set $\mathcal{S}$. Since exclusively sampling from the latent buffer may result in idle actions, we introduce an exploration rate $\delta$, inspired by the $\epsilon$-greedy method from Reinforcement Learning (Sutton & Barto, 2018). With probability $\delta$, we sample from the standard Gaussian distribution instead, ensuring more diverse behavior.

### 3.3. Implementation Details

In Algorithm 1, we present the pseudocode of Falcon, including threshold $\epsilon$, exploration rate $\delta$ and latent buffer $\mathcal{B}$.

In practice, we can't store all the partial denoising action generated from historical observations because of the limited GPU memory. Considering that earlier actions are generally less relevant to the current decision and that we aim to maximize efficiency, we designed a priority queue to implement the latent buffer. Partial denoising actions $\boldsymbol{a}_{\tau:\tau+T_p}^k$ with earlier timesteps $\tau$ and higher noise level $k$ are given priority for removal from the queue. To prevent Falcon from repeating previous actions, we introduce $k_{\min}$ and filter out partial denoised actions with $k < k_{\min}$ during selection. This ensures that only actions with sufficient noise levels are used in the iterative sampling process.

At timestep $t = 1$, where no prior information is available, Falcon directly samples $\boldsymbol{a}_{t:t+T_p}^K$ from a Gaussian distribution $\mathcal{N}(\mathbf{0}, \mathbf{I})$, and iteratively samples conditional on $\mathbf{O}_t$ by Eq. 11 like DDPM (Ho et al., 2020) Line 7. During this process, Falcon iteratively performs denoising and stores resulting partial denoised actions $\boldsymbol{a}_{t:t+T_p}^k$ in a latent buffer $\mathcal{B}$ on Line 5. This latent buffer records all partial denoised actions generated during the sampling process.

$$
\begin{aligned}
\boldsymbol{a}_{t:t+T_p}^{k-1} &= \frac{1}{\sqrt{\alpha_k}} \left( \boldsymbol{a}_{t:t+T_p}^k - C \right) + \sigma_k \boldsymbol{z}, \\
C &= \frac{1 - \alpha_k}{\sqrt{1 - \bar{\alpha}_k}} \epsilon_\theta(\mathbf{O}_t, \boldsymbol{a}_{t:t+T_p}^k, k), \\
\boldsymbol{z} &\sim \mathcal{N}(\mathbf{0}, \mathbf{I})
\end{aligned} \tag{11}
$$

For timestep $t > 1$, Falcon utilizes one-step estimation according to Eq. 9 on all stored samples $\boldsymbol{a}_{\tau:\tau+T_p}^{k_i}$ in the latent buffer $\mathcal{B}$ conditioned on the current observation $\mathbf{O}_t$, which is the most compute-intensive part of the algorithm but can be efficiently parallelized. Line 13 obtains the partial

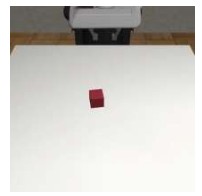 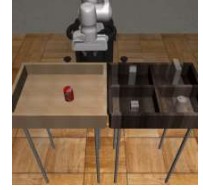 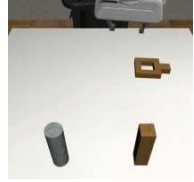 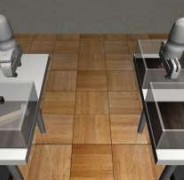 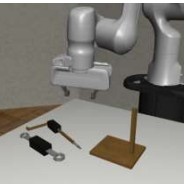

| | Lift | | Can | | Square | | Transport | | ToolHang |
|---|---|---|---|---|---|---|---|---|---|
| | ph | mh | ph | mh | ph | mh | ph | mh | ph |
| DDPM | $1.00_{\pm 0.00}$ | $0.95_{\pm 0.07}$ | $0.98_{\pm 0.13}$ | $0.97_{\pm 0.15}$ | $0.91_{\pm 0.07}$ | $0.85_{\pm 0.35}$ | $0.80_{\pm 0.39}$ | $0.65_{\pm 0.47}$ | $0.52_{\pm 0.49}$ |
| DDPM+Falcon | $1.00_{\pm 0.00}$ | $0.97_{\pm 0.18}$ | $0.97_{\pm 0.17}$ | $0.97_{\pm 0.17}$ | $0.95_{\pm 0.23}$ | $0.82_{\pm 0.38}$ | $0.85_{\pm 0.36}$ | $0.66_{\pm 0.48}$ | $0.55_{\pm 0.50}$ |
| SDP(DDPM) | $1.00_{\pm 0.00}$ | $0.98_{\pm 0.12}$ | $0.97_{\pm 0.17}$ | $0.94_{\pm 0.24}$ | $0.88_{\pm 0.32}$ | $0.80_{\pm 0.40}$ | $0.81_{\pm 0.39}$ | $0.46_{\pm 0.49}$ | $0.58_{\pm 0.50}$ |
| DDIM | $1.00_{\pm 0.00}$ | $1.00_{\pm 0.00}$ | $0.99_{\pm 0.07}$ | $0.97_{\pm 0.15}$ | $0.92_{\pm 0.26}$ | $0.87_{\pm 0.33}$ | $0.79_{\pm 0.40}$ | $0.63_{\pm 0.48}$ | $0.55_{\pm 0.50}$ |
| DDIM+Falcon | $1.00_{\pm 0.00}$ | $1.00_{\pm 0.00}$ | $1.00_{\pm 1.00}$ | $0.98_{\pm 0.14}$ | $0.91_{\pm 0.28}$ | $0.85_{\pm 0.36}$ | $0.81_{\pm 0.39}$ | $0.63_{\pm 0.48}$ | $0.54_{\pm 0.50}$ |
| DPMSolver | $0.98_{\pm 0.12}$ | $0.95_{\pm 0.20}$ | $0.97_{\pm 0.21}$ | $0.97_{\pm 0.17}$ | $0.93_{\pm 0.25}$ | $0.84_{\pm 0.36}$ | $0.70_{\pm 0.45}$ | $0.55_{\pm 0.49}$ | $0.58_{\pm 0.49}$ |
| DPMSolver+Falcon | $0.98_{\pm 0.14}$ | $0.96_{\pm 0.20}$ | $0.96_{\pm 0.20}$ | $0.98_{\pm 0.14}$ | $0.94_{\pm 0.24}$ | $0.90_{\pm 0.30}$ | $0.74_{\pm 0.43}$ | $0.54_{\pm 0.50}$ | $0.56_{\pm 0.48}$ |

*Table 1.* **Success Rate in Robomimic.** We present the success rate with 200 evaluation episodes in the format of (mean of success rate) $\pm$ (standard deviation of success rate). Our results show that Falcon matches the original methods.

denoising action sequence, and Line 14 sets it as the starting point of the rest of the iterative sampling process.

Falcon can also be compatible with other diffusion policy acceleration techniques by replacing the solver in Line 7 and Line 18 with other SDE/ODE solvers like DDIM (Song et al., a) and DPMSolver (Lu et al., 2022). The detailed pseudocode is in the Appendix B.

---

**Algorithm 1** Falcon

**Require:** Diffusion model $\epsilon_\theta$ with noise scheduler $\bar{\alpha}_k$, variance $\sigma_k^2$, threshold $\epsilon$, exploration probability $\delta$, latest $T_o$ observations $\mathbf{O}_t$, latent buffer $\mathcal{B}$, temperature scaling factor $\kappa$.

1: **for** $t = 1, \dots, T$ **do**
2:    **if** $t = 1$ **then**
3:      $\boldsymbol{a}_{t:t+T_p}^K \sim \mathcal{N}(\mathbf{0}, \boldsymbol{I})$
4:      **for** $k = K, \dots, 1$ **do**
5:        $\mathcal{B} \leftarrow \mathcal{B} \cup \{\boldsymbol{a}_{t:t+T_p}^k\}$.
6:        $\boldsymbol{z} \sim \mathcal{N}(\mathbf{0}, \boldsymbol{I})$ if $k > 1$, else $\boldsymbol{z} \leftarrow \mathbf{0}$
7:        Sample $\boldsymbol{a}_{t:t+T_p}^{k-1}$ according to Eq. 11
8:      **end for**
9:    **end if**
10:    **if** $t > 1$ **then**
11:      Compute one-step estimation $\hat{\boldsymbol{a}}_{\tau:\tau-T_a+T_p}^{k_i}$ via Eq. 9.
12:      $\mathcal{S} \leftarrow \{\boldsymbol{a}_{\tau:\tau+T_p}^k : \|\hat{\boldsymbol{a}}_{\tau:\tau-T_a+T_p}^{k_i} - \tilde{\boldsymbol{a}}_{\tau:\tau-T_a+T_p}^{k_i}\|_2 < \epsilon\}$
13:      Sample $\boldsymbol{a}_{\tau:\tau+T_p}^{k_s}$ according to Eq. 10
14:      $\boldsymbol{a}_{t:t+T_p}^{k_s} \leftarrow \boldsymbol{a}_{\tau:\tau+T_p}^{k_s}$
15:      **for** $k = k_s, \dots, 1$ **do**
16:        $\mathcal{B} \leftarrow \mathcal{B} \cup \{\boldsymbol{a}_{t:t+T_p}^k\}$.
17:        $\boldsymbol{z} \sim \mathcal{N}(\mathbf{0}, \boldsymbol{I})$ if $k > 1$, else $\boldsymbol{z} \leftarrow \mathbf{0}$
18:        Sample $\boldsymbol{a}_{t:t+T_p}^{k-1}$ according to Eq. 11
19:      **end for**
20:    **end if**
21: **end for**

---

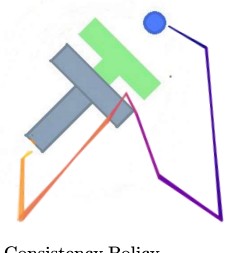 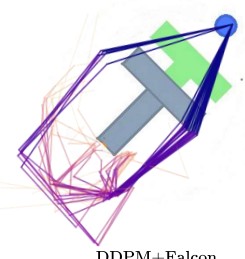

*Figure 3.* **Multimodal behavior.** At the given symmetric configuration in the PushT task, DDPM+Falcon (right) preserves multimodal behaviors by producing diverse reaching trajectories. In contrast, the Consistency Policy (left) shows bias toward one mode. Each line represents one rollout trajectory, plotted from 20 trajectories sampled at the same state.

## 4. Experiments

Our experiments aim to address four key questions: (1) Can Falcon accelerate diffusion policy, and does it further enhance speed when integrated with other acceleration algorithms (Section 4.3)? (2) Can Falcon effectively accelerate diffusion policies in real-world robotic settings while preserving performance (Section 4.5)? (3) Does Falcon maintain its acceleration advantage in long-sequence tasks (Section E.3)? (4) Can Falcon retain the ability to express multimodality while achieving speed improvements (Section 4.6)?

### 4.1. Metrics

Our experiments are primarily evaluated by two metrics. The first is the success rate, which measures the mean and

|  | Lift | | Can | | Square | | Transport | | ToolHang |
|---|---|---|---|---|---|---|---|---|---|
|  | ph | mh | ph | mh | ph | mh | ph | mh | ph |
| DDPM | $100.0_{\pm 0.0}$ | $100.0_{\pm 0.0}$ | $100.0_{\pm 0.0}$ | $100.0_{\pm 0.0}$ | $100.0_{\pm 0.0}$ | $100.0_{\pm 0.0}$ | $100.0_{\pm 0.0}$ | $100.0_{\pm 0.0}$ | $100.0_{\pm 0.0}$ |
| DDPM+Falcon | $\mathbf{12.9}_{\pm 0.3}$ | $\mathbf{24.6}_{\pm 1.6}$ | $\mathbf{35.2}_{\pm 1.3}$ | $\mathbf{20.3}_{\pm 0.3}$ | $\mathbf{37.1}_{\pm 1.5}$ | $\mathbf{20.5}_{\pm 0.6}$ | $\mathbf{46.9}_{\pm 3.7}$ | $\mathbf{48.0}_{\pm 4.7}$ | $\mathbf{33.6}_{\pm 3.2}$ |
| SDP(DDPM) | $50.0_{\pm 0.0}$ | $50.0_{\pm 0.0}$ | $50.0_{\pm 0.0}$ | $50.0_{\pm 0.0}$ | $50.0_{\pm 0.0}$ | $50.0_{\pm 0.0}$ | $50.0_{\pm 0.0}$ | $50.0_{\pm 0.0}$ | $50.0_{\pm 0.0}$ |
| DDIM | $16.0_{\pm 0.0}$ | $16.0_{\pm 0.0}$ | $16.0_{\pm 0.0}$ | $16.0_{\pm 0.0}$ | $16.0_{\pm 0.0}$ | $16.0_{\pm 0.0}$ | $16.0_{\pm 0.0}$ | $16.0_{\pm 0.0}$ | $16.0_{\pm 0.0}$ |
| DDIM+Falcon | $\mathbf{7.5}_{\pm 0.2}$ | $\mathbf{7.7}_{\pm 0.2}$ | $\mathbf{6.5}_{\pm 0.3}$ | $\mathbf{7.6}_{\pm 2.1}$ | $\mathbf{7.6}_{\pm 1.0}$ | $\mathbf{7.2}_{\pm 1.2}$ | $\mathbf{10.5}_{\pm 2.0}$ | $\mathbf{9.0}_{\pm 1.1}$ | $\mathbf{10.1}_{\pm 1.4}$ |
| DPMSolver | $16.0_{\pm 0.0}$ | $16.0_{\pm 0.0}$ | $16.0_{\pm 0.0}$ | $16.0_{\pm 0.0}$ | $16.0_{\pm 0.0}$ | $16.0_{\pm 0.0}$ | $16.0_{\pm 0.0}$ | $16.0_{\pm 0.0}$ | $16.0_{\pm 0.0}$ |
| DPMSolver+Falcon | $\mathbf{6.4}_{\pm 0.1}$ | $\mathbf{10.4}_{\pm 2.0}$ | $\mathbf{14.7}_{\pm 0.8}$ | $\mathbf{10.9}_{\pm 1.8}$ | $\mathbf{14.4}_{\pm 0.5}$ | $\mathbf{7.8}_{\pm 0.9}$ | $\mathbf{11.0}_{\pm 1.2}$ | $\mathbf{12.8}_{\pm 1.4}$ | $\mathbf{12.1}_{\pm 1.3}$ |

*Table 2.* **NFE in Robomimic.** We present the NFE with 200 evaluation episodes in the format of (mean of NFE) ± (standard deviation of NFE). Our results show that Falcon drastically reduces the denoising steps compared with the original methods.

standard deviation of task completion across all trials. The second metric is generation time, quantified in terms of the Number of Function Evaluations (NFE) (Prasad et al., 2024). Since the inference cost for these models is primarily determined by NFE, and given that the network architectures are kept constant across experiments, NFE serves as a reliable indicator of relative performance. This metric effectively captures the inference cost, unbiased by GPU imbalances, and allows for a fair comparison across methods.

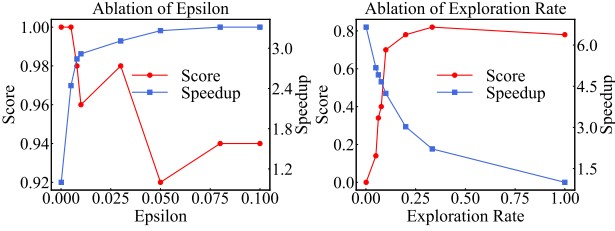

*Figure 4.* **Left**: Ablation analysis on threshold $\epsilon$. The effect of varying the $\epsilon$ threshold on both the score (red circles) and speedup (blue squares), demonstrating the trade-off between performance and speed as $\epsilon$ changes. **Right**: Ablation analysis on exploration rate $\delta$. The score (red circles) increases sharply with $\delta$ and then stabilizes, while the speedup (blue squares) decreases as the exploration rate rises. This study highlights the balance between exploration and exploitation.

### 4.2. Simulation and Real-World Environments

We evaluate Falcon across diverse simulated and real-world environments to demonstrate its generality and effectiveness. Simulation benchmarks include RoboMimic (Mandlekar et al., 2022), Franka Kitchen (Gupta et al., 2020), Block-Push (Shafiullah et al., 2022), PushT (Chi et al., 2023), MetaWorld (Yu et al., 2020), and ManiSkill2 (Gu et al., 2023), covering short- and long-horizon tasks, as well as multimodal behavior tasks. For real-world evaluation, we deploy Falcon on a dual-arm robot platform in two manipulation tasks: dexterous grasping and high-precision insertion. Details of simulation settings, network architectures, and hardware platforms are provided in the Appendix A.

### 4.3. Falcon is a Plug & Play Acceleration Algorithm

In the first experiment, we examine our algorithm on the Robomimic benchmarks. We present the success rates, NFE, and speedup for both the original diffusion models, their Falcon-enhanced counterparts, and SDP in Table 1, Table 2, and Table 4, respectively. To better understand Falcon's impact, we categorize the tasks into simpler and more complex ones based on the level of fine-grained manipulation required. In the Robomimic benchmark, Lift and Can are relatively simple tasks, while the remaining tasks, Square, Transport, and ToolHang, are more complex.

**Simple tasks.** The Lift and Can tasks involve smooth and predictable movements. In Lift, the robotic arm moves downward to grasp a small cube and then upward to lift it, with minimal variation in action within each phase. Can follows a similar pattern, where the robot picks up a coke can from a bin and places it in a target bin, primarily involving steady translational motion. Both tasks exhibit strong dependencies between consecutive actions, which makes them ideal for Falcon acceleration. As shown in Fig. 13 (left), Falcon starts denoising with low noise, significantly speeding up the process. Table 4 shows Falcon achieving a **7x** speedup in Lift and **2-3x** in Can, demonstrating its efficiency in structured motion tasks.

**Difficult tasks.** Square, Transport, and ToolHang are more complex and involve finer, more precise movements. For example, in the Transport task, two robot arms must work together to transfer a hammer from a closed container on a shelf into a target bin on another shelf. One arm retrieves the hammer from the container, while the other arm first clears the target bin by removing trash. Then, one arm hands over the hammer to the other, which must place it in the target bin. These actions are dynamic and entail larger changes between consecutive movements. As a result, Falcon can't start denoising actions with a low noise level (see Fig. 13 (right)). Consequently, Falcon's speedup is not as pronounced as in simpler tasks. However, even in these more challenging scenarios, as shown in Table 4, Falcon still

| | Lift | | Can | | Square | | Transport | | ToolHang |
|---|---|---|---|---|---|---|---|---|---|
| | ph | mh | ph | mh | ph | mh | ph | mh | ph |
| DPMSolver* | $1.00_{\pm 0.00}$ | $0.95_{\pm 0.20}$ | $0.94_{\pm 0.23}$ | $0.96_{\pm 0.20}$ | $0.84_{\pm 0.37}$ ($\downarrow$**10.6%**) | $0.58_{\pm 0.50}$ ($\downarrow$**35.6%**) | $0.69_{\pm 0.46}$ ($\downarrow$**6.8%**) | $0.54_{\pm 0.50}$ | $0.53_{\pm 0.50}$ |
| DPMSolver+Falcon | $0.98_{\pm 0.14}$ | $0.96_{\pm 0.20}$ | $0.96_{\pm 0.20}$ | $0.98_{\pm 0.14}$ | $0.94_{\pm 0.24}$ | $0.90_{\pm 0.30}$ | $0.74_{\pm 0.43}$ | $0.54_{\pm 0.50}$ | $0.56_{\pm 0.48}$ |

*Table 3.* **Reduced-step DPMSolver.** We report the success rate across 200 evaluation rollouts in the format of (mean $\pm$ std). DPMSolver* uses the same number of sampling steps as NFE of DPMSolver(16 steps)+Falcon. Entries marked with $\downarrow$ indicate performance drops compared to DPMSolver(16 steps)+Falcon. The percentage drop is computed as (Falcon score $-$ DPMSolver* score)/Falcon score $\times$ 100%.

| | Lift | | Can | | Square | | Transport | | ToolHang |
|---|---|---|---|---|---|---|---|---|---|
| | ph | mh | ph | mh | ph | mh | ph | mh | ph |
| DDPM+Falcon | $7.78_{\pm 0.19}$ | $4.07_{\pm 0.23}$ | $2.84_{\pm 0.01}$ | $4.93_{\pm 0.06}$ | $2.70_{\pm 0.10}$ | $4.88_{\pm 0.13}$ | $2.14_{\pm 0.13}$ | $2.10_{\pm 0.16}$ | $2.86_{\pm 0.26}$ |
| SDP(DDPM) | $2.00_{\pm 0.00}$ | $2.00_{\pm 0.00}$ | $2.00_{\pm 0.00}$ | $2.00_{\pm 0.00}$ | $2.00_{\pm 0.00}$ | $2.00_{\pm 0.00}$ | $2.00_{\pm 0.00}$ | $2.00_{\pm 0.00}$ | $2.00_{\pm 0.00}$ |
| DDIM+Falcon | $2.13_{\pm 0.03}$ | $2.08_{\pm 0.06}$ | $2.45_{\pm 0.11}$ | $2.15_{\pm 0.24}$ | $2.15_{\pm 0.22}$ | $2.27_{\pm 0.32}$ | $1.57_{\pm 0.24}$ | $1.88_{\pm 0.18}$ | $1.61_{\pm 0.20}$ |
| DPMSolver+Falcon | $2.48_{\pm 0.02}$ | $1.60_{\pm 0.32}$ | $1.09_{\pm 0.07}$ | $1.51_{\pm 0.25}$ | $1.11_{\pm 0.04}$ | $2.09_{\pm 0.21}$ | $1.48_{\pm 0.17}$ | $1.27_{\pm 0.13}$ | $1.32_{\pm 0.12}$ |

*Table 4.* **Speedup in Robomimic.** We present the speed with 200 evaluation episodes in the format of (mean of speedup) $\pm$ (standard deviation of speedup). Speed for X+Falcon is calculated by NFE of X / NFE of X+Falcon.

achieves around **2x** speedup while maintaining performance close to that of the original models.

We further evaluate Falcon in MetaWorld environments using 3D Diffusion Policy (Ze et al., 2024b). The architecture follows the same setup as 3D Diffusion Policy, with a DDIM scheduler using 10-step discretization. As shown in the Appendix C, Falcon achieves a **3-4×** acceleration on top of DDIM, further demonstrating its effectiveness in accelerating existing diffusion-based policies across different tasks.

### 4.4. Falcon mitigates performance degradation in reduced-step solvers

To further demonstrate Falcon's compatibility with existing acceleration techniques, we compare Falcon-enhanced DPMSolver with a reduced-step DPMSolver baseline (denoted as DPMSolver*), where both use the same number of denoising steps. As shown in Table 3, DPMSolver* exhibits significant performance drops in several tasks when the number of steps is aggressively reduced (e.g., $-10.6\%$ in Square_ph, $-35.6\%$ in Square_mh), while Falcon maintains high success rates with the same number of steps. These results highlight Falcon's ability to recover performance in low-step regimes and demonstrate its practical value as a plug-and-play solution to boost efficiency without sacrificing accuracy.

### 4.5. Real-World Evaluation of Falcon

To validate the practicality of Falcon beyond simulation, we evaluate it in two real-world robotic manipulation tasks: Dexterous Grasping (Fig. 5) and Square Stick Insertion (Fig. 6).

**Dexterous Grasping.** The first task involves dexterous

grasping to pick up the blue object placed on a table. Falcon is deployed on top of DDPM without any additional retraining. We compare Falcon+DDPM against SDP(DDPM) and DDPM alone. As shown in Table 5, Falcon achieves a **3.07×** speedup over DDPM and outperforms SDP in both runtime (0.145s vs. 0.233s) and CPU memory usage (3730MB vs. 3808MB).

**High-Precision Insertion.** The second task involves inserting a square stick into a tall cylindrical chip can. This task requires precise 3D alignment; even slight deviations in height or angle result in failure. Falcon is again applied as a plug-in to DDPM. We train both DDPM and SDP using 50 human demonstrations. As shown in Table 6, Falcon matches DDPM in 90% success rate while being **2.86×** faster and more memory efficient.

These results confirm that Falcon not only accelerates diffusion policy inference in simulation but also effectively transfers to real-world robotic applications.

### 4.6. Falcon retains the ability to express multimodality policy

We further evaluate Falcon in the PushT task under a symmetric configuration, where multimodality can be visualized directly through trajectory diversity. As shown in Fig. 3, DDPM+Falcon produces diverse reaching trajectories, while Consistency Policy exhibits mode collapse, predominantly favoring a single direction. This qualitative difference demonstrates Falcon's ability to maintain diverse outputs even in symmetric settings, without retraining.

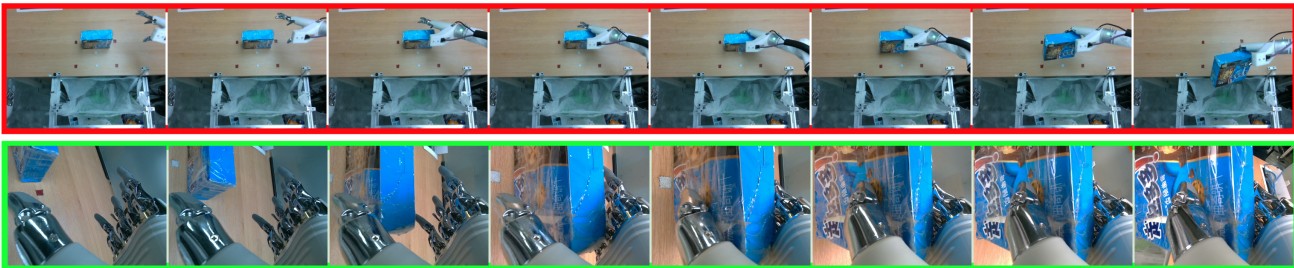

*Figure 5.* **Real-world Dexterous Grasping with DDPM+Falcon.** Top row (red): head camera view (RealSense D435). Bottom row (green): wrist camera view (RealSense D405C). The sequence shows the successful execution of grasping using Falcon-accelerated policy.

| Method | Speedup | NFE | Run Time (s) | GPU MEMS (MB) | CPU MEMS (MB) | Success rate |
|---|---|---|---|---|---|---|
| DDPM | 1.0x | $100.00 \pm 0.00$ | $0.43 \pm 0.01$ | $3730.17 \pm 0.56$ | $3737.06 \pm 2.63$ | 100% |
| SDP(DDPM) | 1.86x | $52.16 \pm 2.18$ | $0.23 \pm 0.01$ | $3733.11 \pm 2.68$ | $3808.67 \pm 7.68$ | 100% |
| Falcon+DDPM | **3.07x** | $30.98 \pm 1.33$ | $0.14 \pm 0.01$ | $3730.19 \pm 0.56$ | $3746.67 \pm 3.03$ | 100% |

*Table 5.* **Dexterous Grasping in Real-world Environment.** Each entry is evaluated with 20 rollouts in the mean $\pm$ standard deviation format. DDPM and SDP(DDPM) are trained with 1000 epochs with 37 collected expert demonstrations. Our results show that Falcon outperforms SDP in runtime reduction with lower memory usage.

### 4.7. Ablation analysis: sensitivity to the threshold $\epsilon$

We explore how the threshold $\epsilon$ influences Falcon's performance in the Square_ph task. The threshold $\epsilon$ determines the confidence in reusing partial denoised actions toward the desired action $\boldsymbol{a}_{t:t+T_p}$ guided by $\nabla_{\mathbf{A}_t^k} \log p\left(\mathbf{A}_t^k \mid \mathbf{O}_t\right)$. We conducted 50 evaluations using a range of $\epsilon$ values: $\{10^{-4}, 5 \times 10^{-3}, 8 \times 10^{-3}, 10^{-2}, 3 \times 10^{-2}, 5 \times 10^{-2}, 8 \times 10^{-2}, 10^{-1}\}$. As shown in Fig. 4 (left), when $\epsilon$ is small, Falcon is highly selective about using partial denoised actions, leading to slow denoising because Falcon frequently samples from the standard normal distribution. This results in no significant improvement. When $\epsilon$ is large, Falcon becomes overly confident in the partial denoised actions, leading to selecting many incorrect actions that do not align with $\boldsymbol{a}_{t+T_a:t+T_p}$, which causes a sharp drop in Score. The optimal range of $\epsilon$ lies in the middle, where Falcon can achieve speedup without sacrificing task performance, demonstrating that a balanced $\epsilon$ allows Falcon to maintain both efficiency and accuracy.

### 4.8. Ablation analysis: sensitivity to exploration rate $\delta$

We assess the impact of the exploration rate $\delta$ on Falcon by running 50 evaluations on the Transport_ph task, using $\delta \in \{0.001, 0.05, 0.0625, 0.076, 0.1, 0.2, 0.33, 1\}$. Exploration Rate $\delta$ controls the probability that Falcon starts denoising from $\mathcal{N}(\mathbf{0}, \mathbf{I})$ rather than the partial denoised action. This exploration mechanism allows Falcon to fill the latent buffer with more partial denoised actions, increasing exploration and preventing over-reliance on the reference action. As shown in Fig. 4 (right), when $\delta$ is set to 0, Falcon primarily aligns with the reference action at each step. Since the

reference action is not perfectly aligned with the desired action $\mathbf{A}_t$, errors accumulate over time, causing Falcon to generate incorrect actions, resulting in a significant drop in score. As $\delta$ increases, Falcon samples more diverse actions, improving score by reducing the accumulation of errors. However, further increasing $\delta$ results in a decrease in speed, as the model becomes less reliant on the history of denoised actions and starts exploring more random samples from the normal distribution.

### 4.9. Ablation analysis: sensitivity to selection mechanism

We evaluate Falcon's partial denoised action selection mechanism. Falcon adaptively selects partial denoised actions based on $\epsilon$ and the one-step estimation, estimating whether each action can be denoised to $A_t$. We compare this with the case where Falcon is fixed to always sample from actions with noise levels of $K/2$ or $K/5$ (represented by the red and green bars), without considering the adaptive selection mechanism. Fig. 7 (left) shows a significant decrease in Score when Falcon is restricted to these fixed actions, demonstrating that not leveraging the selection mechanism compromises performance. This confirms that Falcon's Selection Mechanism is crucial for ensuring accurate action choices, allowing the model to maintain high performance while accelerating the denoising process.

## 5. Related Work

Diffusion policies are widely used for modeling complex behaviors in robotics. 3D Diffusion Policy (Ze et al., 2024b;

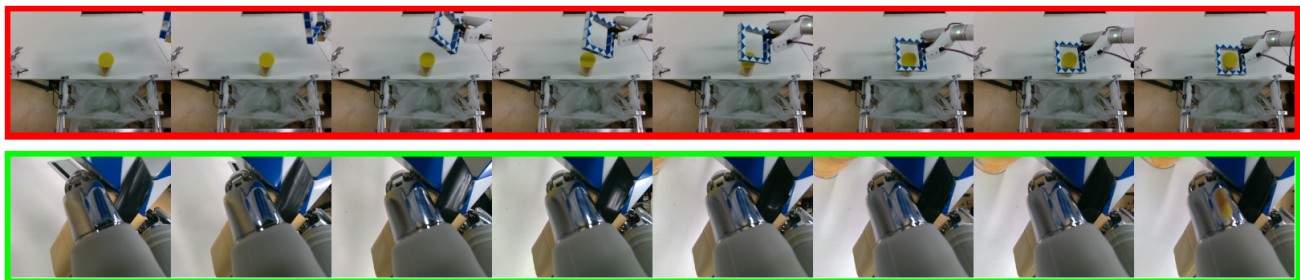

*Figure 6.* **Real-world Square Stick Insertion with DDPM+Falcon.** Top row (red): head camera view (RealSense D435). Bottom row (green): wrist camera view (RealSense D405C). The sequence shows the successful execution of insertion using Falcon-accelerated policy.

| Method | Speedup | NFE | Run Time (s) | GPU MEMS (MB) | Success rate |
|---|---|---|---|---|---|
| DDPM | 1.0x | $100.00_{\pm 0.00}$ | $0.43_{\pm 0.01}$ | $3735.76_{\pm 0.48}$ | 90% |
| SDP(DDPM) | 1.95x | $50.00_{\pm 0.00}$ | $0.22_{\pm 0.01}$ | $3743.28_{\pm 1.26}$ | 85% |
| Falcon+DDPM | **2.86x** | $25.57_{\pm 7.10}$ | $0.15_{\pm 0.12}$ | $3731.50_{\pm 5.08}$ | 90% |

*Table 6.* **Square Stick Insertion in Real-world Environment.** Each entry is evaluated with 20 rollouts in the mean $\pm$ standard deviation format. Falcon is set with $T_p = 32, T_a = 16, T_o = 1, \epsilon = 0.02, |\mathcal{B}| = 20$.

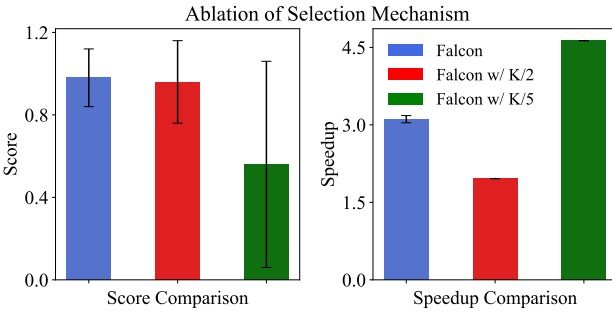

*Figure 7.* Ablation analysis on selection mechanism. This figure compares Falcon's performance with different partial denoised action selection configurations. Restricting Falcon to choose actions with fixed noise levels (K/2, K/5) significantly reduces score, highlighting that the selection mechanism ensures accurate action choices while accelerating denoising and maintaining high performance.

Ke et al.) improves visuomotor performance by integrating 3D visual data from sparse point clouds. Large-scale models like Octo (Team et al., 2024) and RDT (Liu et al., 2024) scale diffusion policy parameters to build vision-language-action foundation models. However, their high computational cost limits real-time applications, necessitating acceleration techniques.

To speed up inference, ODE-based sampling methods like DDIM (Song et al., a) and DPMSolver (Lu et al., 2022) reduce denoising steps, while ParaDiGMS (Shih et al., 2024) leverages parallel computing to accelerate sampling on GPUs. Distillation-based methods further improve efficiency by training models for single-step inference. CP (Prasad et al., 2024) builds on the Consistency Trajectory Model (Kim et al.), enabling a pre-trained diffusion policy to generate actions within a few steps. ManiCM (Lu et al., 2024) extends this approach to 3D robotic tasks. OneDP (Wang et al., 2024) and SDM (Jia et al., 2024) introduce score-based distillation to reduce performance degradation. However, these methods require task-specific training, limiting adaptability. Streaming Diffusion Policy (SDP) (Høeg et al., 2024) uses partial denoised action trajectories, similar to Falcon. However, SDP requires task-specific training and lacks compatibility with general acceleration techniques.

In contrast, Falcon is a training-free acceleration framework that leverages sequential dependencies to improve sampling speed while preserving multimodal expressiveness. It integrates seamlessly with DDIM and DPMSolver, making it a flexible, plug-and-play solution across various robotic tasks.

## 6. Conclusion

This paper introduces Falcon, the first diffusion policy approach to accelerate action generation in complex visuomotor tasks by leveraging inter-step dependencies in decision-making. Empirical results confirm that Falcon outperforms strong baselines, such as DDIM and DPMSolver, in terms of speed without sacrificing accuracy or expressiveness. Overall, Falcon offers a simple yet effective solution for real-time robotic tasks, providing efficient action generation for complex visuomotor environments. However, one limitation of Falcon is that it does not use a single set of parameters to accelerate all tasks, since different environments may require task-specific parameter tuning. Future work will explore adaptive parameter selection strategies to enhance its generalizability across diverse robotic applications further.

## Impact Statement

Falcon removes the efficiency bottleneck of diffusion-based visuomotor policies, making real-time action generation feasible for robotic manipulation, autonomous navigation, and interactive AI. By enabling fast, high-quality decision-making in long-horizon tasks, Falcon expands the practical use of diffusion policies in real-world robot applications.

## Acknowledgements

This work is sponsored by National Natural Science Foundation of China (62376013). We sincerely thank Tianjie He for his invaluable support in setting up the hardware platform used in our real-world experiments.

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

# A. Experimental Setup

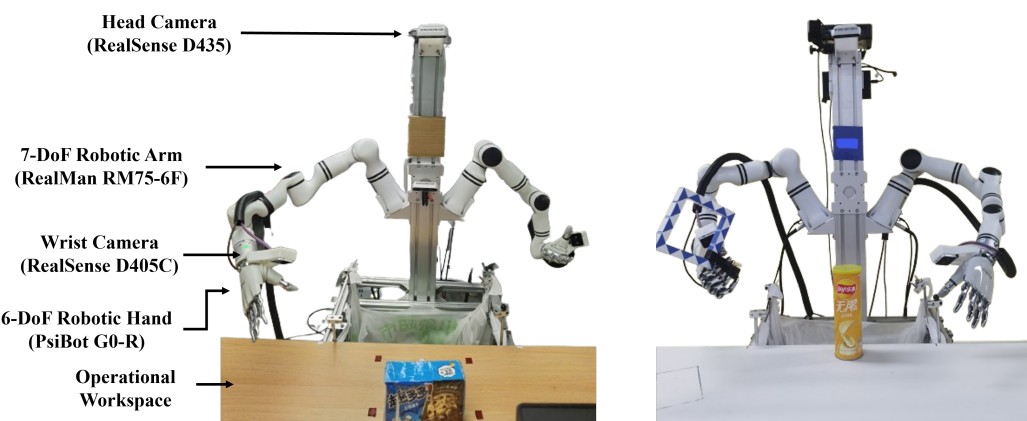

*Figure 8.* **The hardware platform.** Left: Task for Dexterous Grasping. Right: Task for inserting a stick into a chip can

## A.1. Simulation Environments

**Robomimic** (Mandlekar et al., 2022) is a large-scale benchmark designed to evaluate robotic manipulation algorithms using human demonstration datasets. Each task provides two types of human demonstrations: proficient human (PH) demonstrations and mixed proficient and nonproficient human (MH) demonstrations. Each environment uses an action prediction horizon $T_p = 16$ and an action execution horizon $T_a = 8$, and all tasks use state-based observations. We construct a CNN-based Diffusion Policy (Chi et al., 2023) with DDPM scheduler using 100 denoising steps and the DDIM/DPMSolver scheduler using 16 denoising steps. Models and are taken from the Diffusion Policy repository: https://diffusion-policy.cs.columbia.edu/data/experiments/low_dim/.

**Franka Kitchen** (Gupta et al., 2020) is designed for evaluating algorithms on long-horizon, multi-stage robotics tasks. The task involves action sequences of dimension 112 and an episode length of 1200, with an action prediction horizon of $T_p = 16$ and an action execution horizon of $T_a = 8$. We construct a transformer-based Diffusion Policy using a DDPM scheduler with 100-step discretization.

**Push-T** is adapted from IBC (Florence et al., 2022) and involves pushing a block in the shape of a T to a fixed target using a circular end effector. The task features randomized initial poses of the block and end-effector, requiring contact-rich, precise motion. We use state-based observations.

**Multimodel Block Pushing** (Shafiullah et al., 2022) tests the policy's ability to model multimodal action distributions by pushing two blocks into two squares in any order. The demonstration data is generated by a scripted oracle with access to groundtruth state info. This oracle randomly selects an initial block to push and moves it to a randomly selected square. The remaining block is then pushed into the remaining square.

**MetaWorld** (Yu et al., 2020) is an open-source simulation benchmark for meta-reinforcement learning and multi-task learning. It consists of 50 different robotic manipulation tasks categorized into different difficulty levels ranging from simple to very challenging. We follow the setup and model training in the official 3D Diffusion Policy codebase (Ze et al., 2024b): https://github.com/YanjieZe/3D-Diffusion-Policy.

**ManiSkill2** (Gu et al., 2023) is a benchmark for robotic manipulation with 20 tasks, 2000+ objects, and 4M+ demonstrations. It supports rigid/soft-body tasks and fast visual learning (2000 FPS).

## A.2. Real-world Experiments

We conduct two real-world manipulation tasks—dexterous grasping and high-precision insertion. As shown in Figure 8, our robot consists of a 7-DoF RealMan RM75-6F arm and a 6-DoF PsiBot hand. It is equipped with a wrist-mounted RealSense D405C camera and a head-mounted RealSense D435 camera. Objects are placed on a table in front of the robot for manipulation. For each real-world task, we collect 50 human demonstrations and use the open-source Diffusion Policy implementation https://github.com/real-stanford/diffusion_policy and SDP implementations https://github.com/Streaming-Diffusion-Policy/streaming_diffusion_policy.

# B. Pseudocode of Falcon with other sampling solvers

In this section, we provide the pseudocode for Falcon when integrated with alternative sampling solvers, such as DDIM (Song et al., a) and DPMSolver (Lu et al., 2022).

## B.1. Falcon with DDIM

DDIM's sampling process follows Eq. 12, where $\boldsymbol{x}_k$ represents the sample at noise level $k$, $\alpha_k$ is the noise scheduler and $\epsilon_\theta$ is the noise prediction network. To integrate Falcon with DDIM, we replace $\boldsymbol{x}_k$ with partial denoised action sequence $\boldsymbol{a}_{t:t+T_p}^k$ at time step $t$ with noise level $k$ and substitute $\epsilon_\theta(\boldsymbol{x}_k, k)$ with $\epsilon_\theta\left(\mathbf{O}_t, \boldsymbol{a}_{t:t+T_p}^k, k\right)$ where $\mathbf{O}_t$ is the latest $T_o$ observations. This results in the modified sampling process for Falcon-enhanced diffusion policy (Eq. 13). The corresponding pseudocode is provided in Algorithm 2.

$$\boldsymbol{x}_{k-1} = \sqrt{\alpha_{k-1}}\left(\frac{\boldsymbol{x}_k - \sqrt{1-\alpha_k}\epsilon_\theta(\boldsymbol{x}_k, k)}{\sqrt{\alpha_k}}\right) + \sqrt{1-\alpha_{k-1}-\sigma_k^2}\cdot\epsilon_\theta(\boldsymbol{x}_k, k) + \sigma_k\epsilon_k, \ \epsilon_k \sim \mathcal{N}(\mathbf{0},\mathbf{I}) \quad (12)$$

$$\boldsymbol{a}_{t:t+T_p}^{k-1} = \sqrt{\alpha_{k-1}}\left(\frac{\boldsymbol{a}_{t:t+T_p}^k - \sqrt{1-\alpha_k}\epsilon_\theta\left(\mathbf{O}_t, \boldsymbol{a}_{t:t+T_p}^k, k\right)}{\sqrt{\alpha_k}}\right) + \sqrt{1-\alpha_{k-1}-\sigma_k^2}\cdot\epsilon_\theta\left(\mathbf{O}_t, \boldsymbol{a}_{t:t+T_p}^k, k\right) + \sigma_k\boldsymbol{z} \quad (13)$$

---

**Algorithm 2** Falcon: Fast Visuomotor Policies via Partial Denoising (DDIM)

---

**Require:** Diffusion model $\epsilon_\theta$ with noise scheduler $\bar{\alpha}_k$, variance $\sigma_k^2$, threshold $\epsilon$, exploration probability $\delta$, latest $T_o$ observations $\mathbf{O}_t$, latent buffer $\mathcal{B}$, $M+1$ denoising steps $\{k_i\}_{i=0}^M$.

1: **for** $t = 1, \ldots, T$ **do**
2:    **if** $t = 1$ **then**
3:       $\boldsymbol{a}_{t:t+T_p}^K \sim \mathcal{N}(\mathbf{0},\boldsymbol{I})$
4:       **for** $i = M, \ldots, 1$ **do**
5:          $\mathcal{B} \leftarrow \mathcal{B} \cup \{\boldsymbol{a}_{t:t+T_p}^{k_i}\}$.
6:          $\boldsymbol{z} \sim \mathcal{N}(\mathbf{0},\boldsymbol{I})$ if $k_i > 1$, else $\boldsymbol{z} \leftarrow \mathbf{0}$
7:          $\boldsymbol{a}_{t:t+T_p}^{k_{i-1}} \leftarrow \sqrt{\alpha_{k_{i-1}}}\left(\frac{\boldsymbol{a}_{t:t+T_p}^{k_i} - \sqrt{1-\alpha_{k_i}}\epsilon_\theta\left(\mathbf{O}_t, \boldsymbol{a}_{t:t+T_p}^{k_i}, k_i\right)}{\sqrt{\alpha_{k_i}}}\right) + \sqrt{1-\alpha_{k_{i-1}}-\sigma_{k_{i-1}}^2}\cdot\epsilon_\theta\left(\mathbf{O}_t, \boldsymbol{a}_{t:t+T_p}^{k_i}, k_i\right) + \sigma_{k_i}\boldsymbol{z}$
8:       **end for**
9:    **end if**
10:   **if** $t > 1$ **then**
11:      $\hat{\boldsymbol{a}}_{\tau:\tau-T_a+T_p}^{k_i} \leftarrow \frac{1}{\sqrt{\alpha_{k_i}}}\left(\boldsymbol{a}_{\tau:\tau-T_a+T_p}^{k_i} - \sqrt{1-\bar{\alpha}_{k_i}}\epsilon_\theta(\mathbf{O}_t, \boldsymbol{a}_{\tau:\tau-T_a+T_p}^{k_i}, k_i)\right) \quad \forall \boldsymbol{a}_{\tau:\tau-T_a+T_p}^{k_i} \in \mathcal{B}$
12:      $\mathcal{S} \leftarrow \{\boldsymbol{a}_{\tau:\tau+T_p}^k : \|\hat{\boldsymbol{a}}_{\tau:\tau-T_a+T_p}^k - \tilde{\boldsymbol{a}}_{\tau:\tau-T_a+T_p}^k\|_2 < \epsilon, \forall \tau < t, k \in \{k_i\}_{i=0}^M\}$
13:      Sample $\boldsymbol{a}_{\tau:\tau+T_p}^{k_s}$ according to Eq. 10
14:      $\boldsymbol{a}_{t:t+T_p}^{k_s} \leftarrow \boldsymbol{a}_{\tau:\tau+T_p}^{k_s}$
15:      **for** $i = s, \ldots, 1$ **do**
16:         $\mathcal{B} \leftarrow \mathcal{B} \cup \{\boldsymbol{a}_{t:t+T_p}^{k_i}\}$.
17:         $\boldsymbol{z} \sim \mathcal{N}(\mathbf{0},\boldsymbol{I})$ if $k_i > 1$, else $\boldsymbol{z} \leftarrow \mathbf{0}$
18:         $\boldsymbol{a}_{t:t+T_p}^{k_{i-1}} \leftarrow \sqrt{\alpha_{k_{i-1}}}\left(\frac{\boldsymbol{a}_{t:t+T_p}^{k_i} - \sqrt{1-\alpha_{k_i}}\epsilon_\theta\left(\mathbf{O}_t, \boldsymbol{a}_{t:t+T_p}^{k_i}, k_i\right)}{\sqrt{\alpha_{k_i}}}\right) + \sqrt{1-\alpha_{k_{i-1}}-\sigma_{k_{i-1}}^2}\cdot\epsilon_\theta\left(\mathbf{O}_t, \boldsymbol{a}_{t:t+T_p}^{k_i}, k_i\right) + \sigma_{k_i}\boldsymbol{z}$
19:      **end for**
20:   **end if**
21: **end for**

---

## B.2. Falcon with DPMSolver

Given a noise prediction network $\epsilon_\theta$, denoising steps $\{k_i\}_{i=0}^M$, DPMSolver's sampling process follows Eq. 14, where $h_{i-1} = \log \frac{\alpha_{k_{i-1}}}{\sigma_{k_{i-1}}} - \log \frac{\alpha_{k_i}}{\sigma_{k_i}}$. To integrate Falcon, we replace $\boldsymbol{x}_{k_i}$ with $\boldsymbol{a}_{t:t+T_p}^{k_i}$ and substitute $\epsilon_\theta(\boldsymbol{x}_{k_i}, k_i)$ with $\epsilon_\theta\left(\mathbf{O}_t, \boldsymbol{a}_{t:t+T_p}^{k_i}, k_i\right)$, where $\mathbf{O}_t$ represents the latest $T_o$ observations. This yields the sampling process in diffusion policy with DPMSolver, as expressed in Eq. 15. The corresponding pseudocode is provided in Algorithm 3.

$$\boldsymbol{x}_{k_{i-1}} = \frac{\alpha_{k_{i-1}}}{\alpha_{k_i}} \boldsymbol{x}_{k_i} - \sigma_{k_{i-1}} \left(e^{h_{i-1}} - 1\right) \epsilon_\theta\left(\boldsymbol{x}_{k_i}, k_i\right) \tag{14}$$

$$\boldsymbol{a}_{t:t+T_p}^{k_{i-1}} = \frac{\alpha_{k_{i-1}}}{\alpha_{k_i}} \boldsymbol{a}_{t:t+T_p}^{k_i} - \sigma_{k_{i-1}} \left(e^{h_{i-1}} - 1\right) \epsilon_\theta\left(\mathbf{O}_t, \boldsymbol{a}_{t:t+T_p}^{k_i}, k_i\right) \tag{15}$$

---

**Algorithm 3** Falcon: Fast Visuomotor Policies via Partial Denoising (DPMSolver)

---

**Require:** Diffusion model $\epsilon_\theta$ with noise scheduler $\bar{\alpha}_k$, variance $\sigma_k^2$, threshold $\epsilon$, exploration probability $\delta$, latest $T_o$ observations $\mathbf{O}_t$, latent buffer $\mathcal{B}$ and $M+1$ denoising steps $\{k_i\}_{i=0}^M$.

1: **for** $t = 1, \ldots, T$ **do**
2:     **if** $t = 1$ **then**
3:         $\boldsymbol{a}_{t:t+T_p}^K \sim \mathcal{N}(\mathbf{0}, \boldsymbol{I})$
4:         $\boldsymbol{a}_{t:t+T_p}^{k_M} \leftarrow \boldsymbol{a}_{t:t+T_p}^K$
5:         **for** $i = M, \ldots, 1$ **do**
6:             $\mathcal{B} \leftarrow \mathcal{B} \cup \{\boldsymbol{a}_{t:t+T_p}^{k_i}\}$.
7:             $h_{i-1} \leftarrow \log \frac{\alpha_{k_{i-1}}}{\sigma_{k_{i-1}}} - \log \frac{\alpha_{k_i}}{\sigma_{k_i}}$
8:             $\boldsymbol{a}_{t:t+T_p}^{k_{i-1}} = \frac{\alpha_{k_{i-1}}}{\alpha_{k_i}} \boldsymbol{a}_{t:t+T_p}^{k_i} - \sigma_{k_{i-1}} \left(e^{h_{i-1}} - 1\right) \epsilon_\theta\left(\mathbf{O}_t, \boldsymbol{a}_{t:t+T_p}^{k_i}, k_i\right)$
9:         **end for**
10:     **end if**
11:     **if** $t > 1$ **then**
12:         $\hat{\boldsymbol{a}}_{\tau:\tau-T_a+T_p}^{k_i} \leftarrow \frac{1}{\sqrt{\alpha_{k_i}}} \left(\boldsymbol{a}_{\tau:\tau-T_a+T_p}^{k_i} - \sqrt{1 - \bar{\alpha}_{k_i}} \epsilon_\theta(\mathbf{O}_t, \boldsymbol{a}_{\tau:\tau-T_a+T_p}^{k_i}, k_i)\right) \quad \forall \boldsymbol{a}_{\tau:\tau-T_a+T_p}^{k_i} \in \mathcal{B}$
13:         $\mathcal{S} \leftarrow \{\boldsymbol{a}_{\tau:\tau+T_p}^k : \|\hat{\boldsymbol{a}}_{\tau:\tau-T_a+T_p}^{k_i} - \tilde{\boldsymbol{a}}_{\tau:\tau-T_a+T_p}^{k_i}\|_2 < \epsilon, \forall \tau < t\}$
14:         Sample $\boldsymbol{a}_{\tau:\tau+T_p}^{k_s} \sim P(\boldsymbol{a}_{\tau:\tau+T_p}^{k_s}) = \frac{\exp k_s}{\sum \exp k_i}$ in $\mathcal{S}$
15:         $\boldsymbol{a}_{t:t+T_p}^{k_s} \leftarrow \boldsymbol{a}_{\tau:\tau+T_p}^{k_s}$
16:         **for** $i = s, \ldots, 1$ **do**
17:             $\mathcal{B} \leftarrow \mathcal{B} \cup \{\boldsymbol{a}_{t:t+T_p}^{k_i}\}$.
18:             $h_{i-1} \leftarrow \log \frac{\alpha_{k_{i-1}}}{\sigma_{k_{i-1}}} - \log \frac{\alpha_{k_i}}{\sigma_{k_i}}$
19:             $\boldsymbol{a}_{t:t+T_p}^{k_{i-1}} = \frac{\alpha_{k_{i-1}}}{\alpha_{k_i}} \boldsymbol{a}_{t:t+T_p}^{k_i} - \sigma_{k_{i-1}} \left(e^{h_{i-1}} - 1\right) \epsilon_\theta\left(\mathbf{O}_t, \boldsymbol{a}_{t:t+T_p}^{k_i}, k_i\right)$
20:         **end for**
21:     **end if**
22: **end for**

# C. Detail performance of Falcon with 3D Diffusion Policy

In this section, we provide a detailed results of Falcon's acceleration performance when applied to 3D Diffusion Policy (Ze et al., 2024b) in MetaWorld environments. Falcon is integrated with DDIM using 10-step discretization(we call 3D FalconDDIM), following the original 3D Diffusion Policy architecture to ensure a fair comparison.

Tables 7, 8 and 9 present the success rates, NFE and speedup respectively, for both 3D Diffusion Policy and 3D FalconDDIM. These results further validate Falcon's effectiveness in reducing inference time while maintaining task performance across different robotic manipulation tasks.

*Table 7.* **Detailed results for 39 simulated tasks with success rates.** We evaluated 39 challenging tasks using 50 random seeds and reported the average success rate (%) and standard deviation for each task individually. The 3D FalconDDIM algorithm demonstrates nearly no performance drop.

| Alg \ Task | Meta-World (Easy) | | | | | |
|---|---|---|---|---|---|---|
| | Button Press | Coffee Button | Plate Slide Back Side | Plate Slide Side | Window Close | Window Open |
| 3D Diffusion Policy | $100 \pm 0$ | $100 \pm 0$ | $100 \pm 0$ | $100 \pm 0$ | $100 \pm 0$ | $100 \pm 0$ |
| 3D FalconDDIM | $100 \pm 0$ | $100 \pm 0$ | $100 \pm 0$ | $100 \pm 0$ | $100 \pm 0$ | $100 \pm 0$ |

| Alg \ Task | Meta-World (Easy) | | | | | |
|---|---|---|---|---|---|---|
| | Button Press Topdown | Button Press Topdown Wall | Button Press Wall | Peg Unplug Side | Door Close | Door Lock |
| 3D Diffusion Policy | $100 \pm 0$ | $99 \pm 2$ | $99 \pm 1$ | $75 \pm 5$ | $100 \pm 0$ | $98 \pm 2$ |
| 3D FalconDDIM | $100 \pm 0$ | $100 \pm 0$ | $100 \pm 0$ | $75 \pm 43$ | $100 \pm 0$ | $96 \pm 19$ |

| Alg \ Task | Meta-World (Easy) | | | | | | |
|---|---|---|---|---|---|---|---|
| | Door Open | Door Unlock | Drawer Close | Drawer Open | Faucet Close | Faucet Open | Handle Press |
| 3D Diffusion Policy | $99 \pm 1$ | $100 \pm 0$ | $100 \pm 0$ | $100 \pm 0$ | $100 \pm 0$ | $100 \pm 0$ | $100 \pm 0$ |
| 3D FalconDDIM | $100 \pm 0$ | $100 \pm 0$ | $100 \pm 0$ | $100 \pm 0$ | $100 \pm 0$ | $100 \pm 0$ | $100 \pm 0$ |

| Alg \ Task | Meta-World (Easy) | | | | | |
|---|---|---|---|---|---|---|
| | Handle Pull Side | Lever Pull | Plate Slide | Plate Slide Back | Dial Turn | Reach |
| 3D Diffusion Policy | $85 \pm 3$ | $79 \pm 8$ | $100 \pm 1$ | $99 \pm 0$ | $92 \pm 27$ | $68 \pm 46$ |
| 3D FalconDDIM | $87 \pm 33$ | $81 \pm 39$ | $86 \pm 34$ | $100 \pm 0$ | $89 \pm 31$ | $68 \pm 46$ |

| Alg \ Task | Meta-World (Medium) | | | | | | Meta-World (Hard) | |
|---|---|---|---|---|---|---|---|---|
| | Hammer | Basketball | Push Wall | Box Close | Sweep | Sweep Into | Assembly | Hand Insert |
| 3D Diffusion Policy | $88 \pm 32$ | $98 \pm 2$ | $88 \pm 32$ | $56 \pm 49$ | $96 \pm 3$ | $15 \pm 5$ | $99 \pm 1$ | $12 \pm 32$ |
| 3D FalconDDIM | $83 \pm 37$ | $100 \pm 0$ | $88 \pm 32$ | $55 \pm 49$ | $100 \pm 0$ | $13 \pm 33$ | $99 \pm 9$ | $26 \pm 43$ |

| Alg \ Task | Meta-World (Hard) | Meta-World (Very Hard) | | | | |
|---|---|---|---|---|---|---|
| | Push | Shelf Place | Disassemble | Stick Pull | Stick Push | Pick Place Wall |
| 3D Diffusion Policy | $51 \pm 3$ | $52 \pm 49$ | $72 \pm 44$ | $68 \pm 46$ | $97 \pm 4$ | $80 \pm 40$ |
| 3D FalconDDIM | $53 \pm 49$ | $47 \pm 49$ | $75 \pm 43$ | $68 \pm 46$ | $100 \pm 0$ | $87 \pm 33$ |

*Table 8.* **Detailed results for 39 simulated tasks with NFE.** We evaluated 39 challenging tasks using 50 random seeds and reported the average Number of Function Evaluations (nfe) per action generation and standard deviation for each domain individually. The 3D FalconDDIM algorithm reduces the nfe to a range of 2-4 compared to the 3D Diffusion Policy

| Alg \ Task | Meta-World (Easy) | | | | | |
| --- | --- | --- | --- | --- | --- | --- |
| | Button Press | Coffee Button | Plate Slide Back Side | Plate Slide Side | Window Close | Window Open |
| 3D Diffusion Policy | $10 \pm 0$ | $10 \pm 0$ | $10 \pm 0$ | $10 \pm 0$ | $10 \pm 0$ | $10 \pm 0$ |
| 3D FalconDDIM | $\mathbf{2.12 \pm 0.05}$ | $\mathbf{2.47 \pm 0.09}$ | $\mathbf{3.91 \pm 0.22}$ | $\mathbf{3.27 \pm 0.68}$ | $\mathbf{2.54 \pm 0.18}$ | $\mathbf{3.58 \pm 0.90}$ |

| Alg \ Task | Meta-World (Easy) | | | | | |
| --- | --- | --- | --- | --- | --- | --- |
| | Button Press Topdown | Button Press Topdown Wall | Button Press Wall | Peg Unplug Side | Door Close | Door Lock |
| 3D Diffusion Policy | $10 \pm 0$ | $10 \pm 0$ | $10 \pm 0$ | $10 \pm 0$ | $10 \pm 0$ | $10 \pm 0$ |
| 3D FalconDDIM | $\mathbf{2.48 \pm 0.20}$ | $\mathbf{2.63 \pm 0.40}$ | $\mathbf{3.36 \pm 0.61}$ | $\mathbf{3.25 \pm 0.38}$ | $\mathbf{2.81 \pm 0.05}$ | $\mathbf{4.06 \pm 1.00}$ |

| Alg \ Task | Meta-World (Easy) | | | | | |
| --- | --- | --- | --- | --- | --- | --- |
| | Door Open | Door Unlock | Drawer Close | Drawer Open | Faucet Close | Faucet Open | Handle Press |
| 3D Diffusion Policy | $10 \pm 0$ | $10 \pm 0$ | $10 \pm 0$ | $10 \pm 0$ | $10 \pm 0$ | $10 \pm 0$ | $10 \pm 0$ |
| 3D FalconDDIM | $\mathbf{3.04 \pm 0.26}$ | $\mathbf{4.70 \pm 0.86}$ | $\mathbf{2.91 \pm 0.47}$ | $\mathbf{3.90 \pm 0.18}$ | $\mathbf{4.06 \pm 0.62}$ | $\mathbf{4.94 \pm 0.78}$ | $\mathbf{3.52 \pm 0.43}$ |

| Alg \ Task | Meta-World (Easy) | | | | | |
| --- | --- | --- | --- | --- | --- | --- |
| | Handle Pull Side | Lever Pull | Plate Slide | Plate Slide Back | Dial Turn | Reach |
| 3D Diffusion Policy | $10 \pm 0$ | $10 \pm 0$ | $10 \pm 0$ | $10 \pm 0$ | $10 \pm 0$ | $10 \pm 0$ |
| 3D FalconDDIM | $\mathbf{3.59 \pm 0.51}$ | $\mathbf{3.79 \pm 0.77}$ | $\mathbf{3.11 \pm 0.15}$ | $\mathbf{3.49 \pm 0.16}$ | $\mathbf{4.26 \pm 1.16}$ | $\mathbf{2.85 \pm 0.05}$ |

| Alg \ Task | Meta-World (Medium) | | | | | | Meta-World (Hard) | |
| --- | --- | --- | --- | --- | --- | --- | --- | --- |
| | Hammer | Basketball | Push Wall | Box Close | Sweep | Sweep Into | Assembly | Hand Insert |
| 3D Diffusion Policy | $10 \pm 0$ | $10 \pm 0$ | $10 \pm 0$ | $10 \pm 0$ | $10 \pm 0$ | $10 \pm 0$ | $10 \pm 0$ | $10 \pm 0$ |
| 3D FalconDDIM | $\mathbf{3.25 \pm 0.60}$ | $\mathbf{2.97 \pm 0.14}$ | $\mathbf{3.35 \pm 0.22}$ | $\mathbf{4.51 \pm 1.12}$ | $\mathbf{3.09 \pm 0.18}$ | $\mathbf{3.59 \pm 0.19}$ | $\mathbf{2.44 \pm 0.23}$ | $\mathbf{3.29 \pm 0.19}$ |

| Alg \ Task | Meta-World (Hard) | Meta-World (Very Hard) | | | | |
| --- | --- | --- | --- | --- | --- | --- |
| | Push | Shelf Place | Disassemble | Stick Pull | Stick Push | Pick Place Wall |
| 3D Diffusion Policy | $10 \pm 0$ | $10 \pm 0$ | $10 \pm 0$ | $10 \pm 0$ | $10 \pm 0$ | $10 \pm 0$ |
| 3D FalconDDIM | $\mathbf{2.88 \pm 0.38}$ | $\mathbf{4.28 \pm 2.33}$ | $\mathbf{3.25 \pm 0.37}$ | $\mathbf{3.78 \pm 0.73}$ | $\mathbf{3.41 \pm 0.32}$ | $\mathbf{3.04 \pm 0.15}$ |

*Table 9.* **Detailed results for 39 simulated tasks with speedup.** We evaluated 39 challenging tasks using 50 random seeds and reported the average speedup and standard deviation for each task individually. The 3D FalconDDIM algorithm reduces the NFE to a range of 2-4 compared to the 3D Diffusion Policy

| Alg \ Task | Meta-World (Easy) | | | | | |
|---|---|---|---|---|---|---|
| | Button Press | Coffee Button | Plate Slide Back Side | Plate Slide Side | Window Close | Window Open |
| 3D FalconDDIM | $4.71 \pm 0.12$ | $4.03 \pm 0.14$ | $2.55 \pm 0.14$ | $3.05 \pm 0.60$ | $3.93 \pm 0.28$ | $2.79 \pm 0.66$ |

| Alg \ Task | Meta-World (Easy) | | | | | |
|---|---|---|---|---|---|---|
| | Button Press Topdown | Button Press Topdown Wall | Button Press Wall | Peg Unplug Side | Door Close | Door Lock |
| 3D FalconDDIM | $4.02 \pm 0.32$ | $3.80 \pm 0.48$ | $2.97 \pm 0.47$ | $3.07 \pm 0.34$ | $3.55 \pm 0.06$ | $2.46 \pm 0.51$ |

| Alg \ Task | Meta-World (Easy) | | | | | | |
|---|---|---|---|---|---|---|---|
| | Door Open | Door Unlock | Drawer Close | Drawer Open | Faucet Close | Faucet Open | Handle Press |
| 3D FalconDDIM | $3.28 \pm 0.26$ | $2.12 \pm 0.32$ | $3.42 \pm 0.36$ | $2.55 \pm 0.12$ | $2.45 \pm 0.35$ | $2.02 \pm 0.35$ | $2.83 \pm 0.30$ |

| Alg \ Task | Meta-World (Easy) | | | | | |
|---|---|---|---|---|---|---|
| | Handle Pull Side | Lever Pull | Plate Slide | Plate Slide Back | Dial Turn | Reach |
| 3D FalconDDIM | $2.77 \pm 0.36$ | $2.63 \pm 0.44$ | $3.21 \pm 0.15$ | $2.86 \pm 0.14$ | $2.34 \pm 0.56$ | $3.50 \pm 0.06$ |

| Alg \ Task | Meta-World (Medium) | | | | | | Meta-World (Hard) | |
|---|---|---|---|---|---|---|---|---|
| | Hammer | Basketball | Push Wall | Box Close | Sweep | Sweep Into | Assembly | Hand Insert |
| 3D FalconDDIM | $3.25 \pm 0.60$ | $2.97 \pm 0.14$ | $3.35 \pm 0.22$ | $4.51 \pm 1.12$ | $3.09 \pm 0.18$ | $3.59 \pm 0.19$ | $2.44 \pm 0.23$ | $3.29 \pm 0.19$ |

| Alg \ Task | Meta-World (Hard) | Meta-World (Very Hard) | | | | |
|---|---|---|---|---|---|---|
| | Push | Shelf Place | Disassemble | Stick Pull | Stick Push | Pick Place Wall |
| 3D FalconDDIM | $2.88 \pm 0.38$ | $2.33 \pm 0.65$ | $3.07 \pm 0.30$ | $2.64 \pm 0.44$ | $2.92 \pm 0.25$ | $3.28 \pm 0.14$ |

## D. Experiment Details

In this section, we provide the detailed experimental setup for Robomimic and analyze Falcon's memory cost compared to the original samplers (DDPM, DDIM, and DPMSolver). Table 10 reports the hyperparameter settings and the peak memory usage for each experiment.

To evaluate Falcon's computational overhead, we measure the peak memory cost, denoted in the format (original sampler cost) + (incremental cost due to Falcon integration). As shown in Table 10, Falcon introduces an additional 12 MB of memory overhead, which is negligible compared to the original 1876 MB cost. This demonstrates that Falcon achieves acceleration with minimal memory overhead, making it a practical and efficient enhancement to diffusion-based policies.

| | DDPM+Falcon | | | | | DDIM+Falcon | | | | | DPMSolver+Falcon | | | | |
| --- | --- | --- | --- | --- | --- | --- | --- | --- | --- | --- | --- | --- | --- | --- | --- |
| | $\epsilon$ | $\delta$ | $k_{\min}$ | $|\mathcal{B}|$ | Peak Memory Cost | $\epsilon$ | $\delta$ | $k_{\min}$ | $|\mathcal{B}|$ | Peak Memory Cost | $\epsilon$ | $\delta$ | $k_{\min}$ | $|\mathcal{B}|$ | Peak Memory Cost |
| Lift ph | 0.04 | 0.1 | 20 | 50 | 1876+12 MB | 0.05 | 0.2 | 15 | 50 | 1876+12 MB | 0.08 | 0.2 | 20 | 50 | 1876+12 MB |
| Lift mh | 0.04 | 0.1 | 20 | 50 | 1876+12 MB | 0.03 | 0.25 | 10 | 50 | 1876+12 MB | 0.005 | 0.2 | 20 | 50 | 1876+12 MB |
| Can ph | 0.01 | 0.1 | 20 | 50 | 1876+12 MB | 0.01 | 0.20 | 8 | 50 | 1876+12 MB | 0.003 | 0.2 | 20 | 50 | 1876+12 MB |
| Can mh | 0.01 | 0.1 | 20 | 50 | 1876+12 MB | 0.005 | 0.20 | 8 | 50 | 1876+12 MB | 0.003 | 0.2 | 20 | 50 | 1876+12 MB |
| Square ph | 0.04 | 0.1 | 25 | 50 | 1876+12 MB | 0.01 | 0.20 | 8 | 50 | 1876+12 MB | 0.003 | 0.2 | 25 | 20 | 1876+12 MB |
| Square mh | 0.04 | 0.1 | 25 | 50 | 1876+12 MB | 0.005 | 0.20 | 3 | 50 | 1876+12 MB | 0.005 | 0.2 | 20 | 20 | 1876+12 MB |
| Transport ph | 0.01 | 0.2 | 25 | 50 | 1876+12 MB | 0.001 | 0.33 | 8 | 50 | 1876+12 MB | 0.005 | 0.2 | 20 | 20 | 1876+12 MB |
| Transport mh | 0.01 | 0.2 | 25 | 50 | 1876+12 MB | 0.01 | 0.33 | 5 | 50 | 1876+12 MB | 0.003 | 0.2 | 60 | 50 | 1876+12 MB |
| ToolHang ph | 0.01 | 0.1 | 20 | 50 | 1876+12 MB | 0.003 | 0.33 | 5 | 50 | 1876+12 MB | 0.003 | 0.2 | 60 | 50 | 1876+12 MB |

*Table 10.* **Hyperparameters and Memory Cost in Robomimic.**

# E. Additional Experiments

## E.1. Acceleration on Vision-Language-Action Foundation Model (RDT)

To assess Falcon compatibility with pre-trained diffusion foundation models, we integrate it with RDT-1B (Liu et al., 2024), a vision language action foundation model trained in ManiSkill2. Falcon is applied directly to RDT's DPMSolver++ inference without retraining. The official model weights are obtained from the RDT repository on https://huggingface.co/robotics-diffusion-transformer/maniskill-model.

As shown in Table 11, Falcon achieves substantial inference speedups (31x and 34x vs. DDPM) with no loss in success rate on task PickCube (see Fig. 9) and task PushCube (see Fig. 10), demonstrating its plug-and-play compatibility with VLA Foundation model.

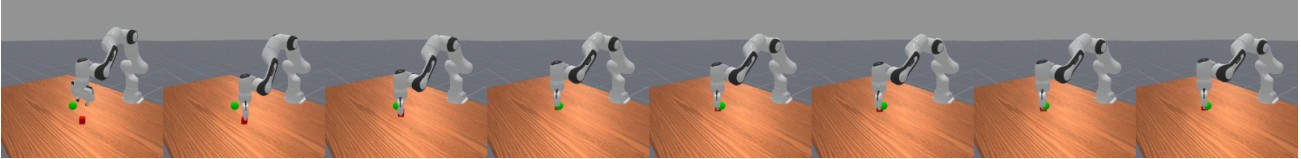

*Figure 9.* RDT evaluated in PickCube with Falcon

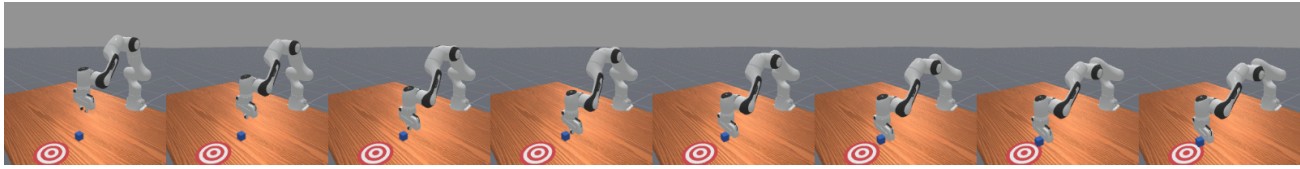

*Figure 10.* RDT evaluated in PushCube with Falcon

| PickCube | | | | | |
| Method | Success Rate | NFE | Time (s) | Speedup | GPU MEMS (MB) |
| --- | --- | --- | --- | --- | --- |
| DPMSolver++ (5 steps) | 0.75 | 5.00 | 0.08 | 20x | 15479 |
| DPMSolver++ (5 steps)+Falcon | 0.75 | 3.22 | 0.04 | 31x | 15483 |

| PushCube | | | | | |
| Method | Success Rate | NFE | Time (s) | Speedup | GPU MEMS (MB) |
| --- | --- | --- | --- | --- | --- |
| DDPMSolver++ (5 steps) | 1.00 | 5.00 | 0.08 | 20x | 15483 |
| DPMSolver++ (5 steps)+Falcon | 1.00 | 2.91 | 0.05 | 34x | 15483 |

*Table 11.* Falcon improves inference speed for RDT on unseen ManiSkill tasks ($T_p = 64, T_a = 32, T_o = 2, \epsilon = 0.02, |\mathcal{B}| = 2$) using the same pre-trained checkpoint. Speedup is relative to 100-step DDPM. Falcon achieves comparable performance with significantly fewer denoising steps and lower runtime.

### E.2. Additional Ablation on Threshold $\epsilon$

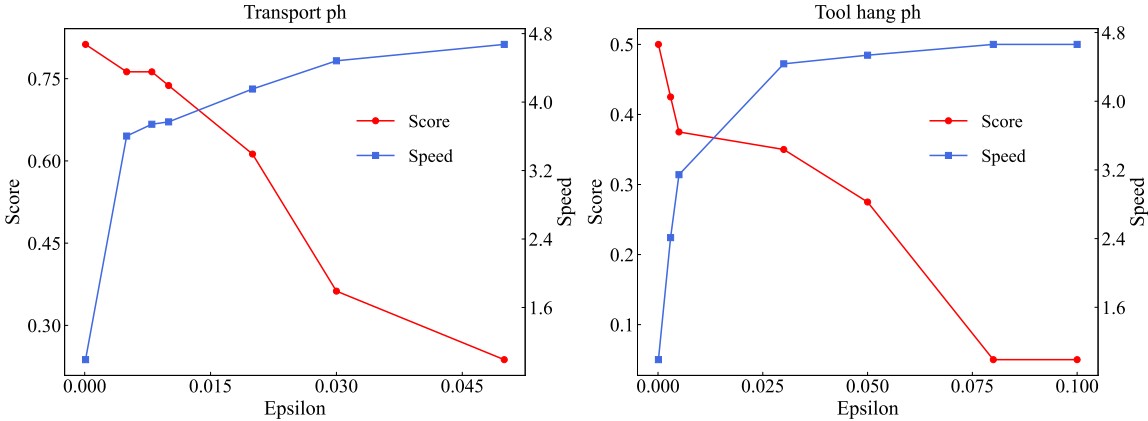

*Figure 11.* **Additional Ablation Analysis on threshold $\epsilon$.** We conduct extended experiments with DDPM+Falcon on *Transport ph* and *Tool hang ph*, varying the threshold $\epsilon$. Results show a trade-off between success rate (Score) and speedup.

### E.3. Falcon achieves accelerating diffusion policy in long-horizon tasks

To evaluate Falcon's effectiveness in long-horizon tasks, we apply it to the Franka Kitchen environment. As shown in Table 12, Falcon maintains the same high success rates as the original DDPM model, confirming that the acceleration introduced by Falcon does not compromise task performance, even for long-horizon tasks. The key benefit of Falcon lies in its ability to reduce NFE and accelerate the denoising process, which is evident in the speedup shown in Table 13. Falcon achieves a **5.25x** speedup in Kitchen, which shows that Falcon can accelerate DDPM in long-horizon tasks and can be adopted into transformer architecture.

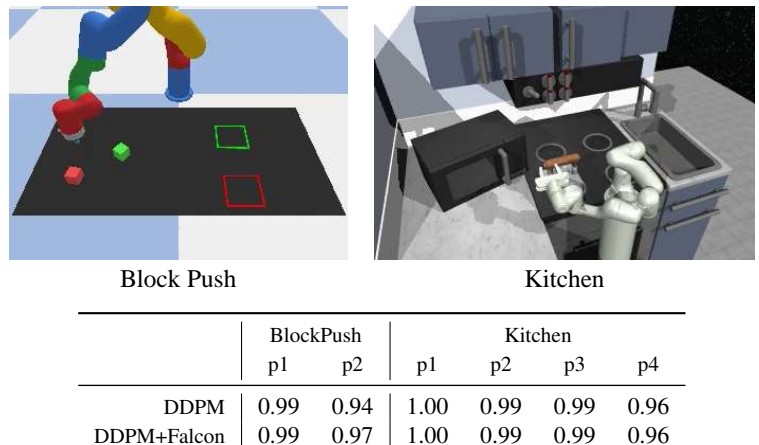

Block Push          Kitchen

|  | BlockPush | | Kitchen | | | |
|---|---|---|---|---|---|---|
|  | p1 | p2 | p1 | p2 | p3 | p4 |
| DDPM | 0.99 | 0.94 | 1.00 | 0.99 | 0.99 | 0.96 |
| DDPM+Falcon | 0.99 | 0.97 | 1.00 | 0.99 | 0.99 | 0.96 |

*Table 12.* **Success Rate in BlockPush and Kitchen.** For BlockPush, p$x$ refers to the frequency of pushing $x$ blocks into the targets. For Kitchen, p$x$ refers to the frequency of interacting with $x$ or more objects.

|  | BlockPush | | Kitchen | |
|---|---|---|---|---|
|  | NFE | Speedup | NFE | Speedup |
| DDPM | 100 | 1.0x | 100 | 1.0x |
| DDPM+Falcon | 32.7 | **2.8x** | 19.03 | **5.25x** |

*Table 13.* **Speedup in BlockPush and Kitchen.** Falcon can accelerate diffusion policy in long-horizon tasks.

## E.4. More results of Multimodality

In this section, we provide a visualization of the multimodal action distributions generated by Falcon in the BlockPush environment. As shown in Fig. 12, the frequency distribution of different action modalities in the BlockPush task, where each modality corresponds to a distinct combination of blocks being pushed into two squares. The chart illustrates the uniformity of the action modality frequencies, with the four modalities (1-2, 1-1, 2-1, and 2-2) being equally represented. This visualization confirms that Falcon is capable of expressing multimodal actions, effectively handling different action combinations without bias.

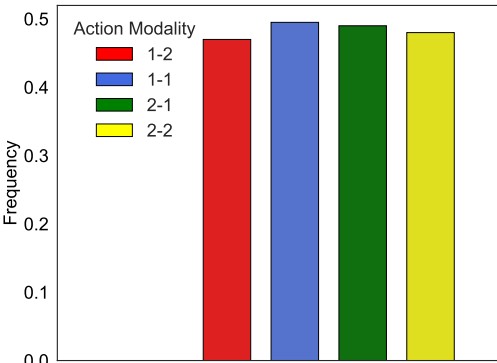

*Figure 12.* **Action Modalities Distribution in BlockPush Task.** The bar chart shows the frequency of different policy modalities for pushing two blocks into two squares in any order. The modalities are represented as 1-2, 1-1, 2-1, and 2-2, where the numbers indicate the block number and square number respectively. The chart illustrates the frequency distribution of the action modalities generated by Falcon in the BlockPush task.

# F. Visualization of Starting Points

This section visualizes where Falcon starts the denoising process at each time step, specifically from which past time step and at what noise level the partial denoised actions originate. We analyze two tasks: Lift ph and Transport ph, representing high and low acceleration scenarios, respectively.

As shown in Fig.13 (left), in the Lift task, Falcon consistently starts denoising from the partial denoised action of the previous time step, with a low noise level. This indicates a significant reduction in sampling steps, leading to substantial acceleration. In contrast, Fig.13 (right) shows that the Transport task starts from actions with higher noise levels, limiting the acceleration effect. This suggests that Falcon's speedup is more pronounced in tasks with smoother, more predictable action transitions.

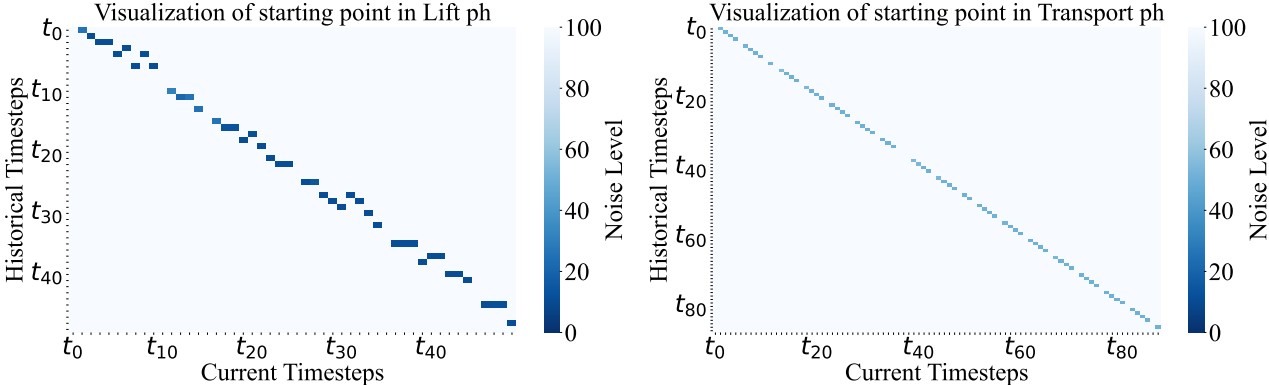

*Figure 13.* **Visualization of denoising starting point.** This figure shows the heatmaps $A \in \mathbb{R}^{T \times T}$ in Lift ph task (left) and Transport ph task (right). $A[j, i] = k$ means that at time step $i$, Falcon starts denoising at the partial denoised actions $\boldsymbol{a}_{j:j+T_p}^{k}$.

