# OpenReview forum: "Falcon: Fast Visuomotor Policies via Partial Denoising"
_ICML.cc/2025/Conference — ICML 2025 poster_

### Official Review · Reviewer_8uiC · 2025-03-10

**Overall Recommendation:** 3

**Summary:**

The paper presents Falcon, an innovative approach that accelerates diffusion-based visuomotor policies without sacrificing their performance. Conventional diffusion policies rely on multiple denoising steps, which can hinder real-time decision-making. Falcon addresses this issue by exploiting the sequential dependencies among actions, enabling the denoising process to start from partially denoised actions rather than from a standard normal distribution. This approach reduces the number of required sampling steps, improving inference speed without the need for extra training.

**Claims And Evidence:**

Overall, the claims made in the Falcon paper are clearly articulated and largely supported by experimental evidence provided by the authors. The paper's main claims revolve around its ability to achieve accelerated inference speed through partial denoising, while maintaining performance and multimodal expressiveness.

The author suggest Falcon is promising for real-world robotics. However, all validations are conducted exclusively in simulation environments. Real-world robot experiments are absent.

**Essential References Not Discussed:**

Yes, there is an important related work that the Falcon paper did not mention, namely the "Streaming Diffusion Policy (SDP)" proposed in the paper titled "Fast Policy Synthesis with Variable Noise Diffusion Models".

SDP also proposes to accelerate diffusion-based policy generation through partial denoising, sharing conceptual similarities with Falcon. Specifically, SDP leverages the insight that partially denoised action trajectories can be generated significantly faster, enabling efficient inference without a large reduction in policy quality. Importantly, SDP is validated in both simulated and real-world robotic tasks, clearly demonstrating its effectiveness in realistic conditions.

**Experimental Designs Or Analyses:**

Overall, the experimental designs and analyses are sound and valid, and align well with standard practice in evaluating diffusion-based visuomotor policies. The chosen benchmark environments and evaluation criteria are appropriate. Additionally, the baseline methods (DDPM, DDIM, DPMSolver) are well-selected and relevant, ensuring fair comparisons.

There are some limitations:

Experiments were conducted exclusively in simulated environments, leaving uncertainty regarding real-world applicability and robustness.

Computational efficiency is primarily measured by NFE, with no explicit runtime or memory consumption data provided.

The proposed method relies on hyperparameters, requiring task-specific tuning for optimal results, potentially complicating practical adoption.

**Methods And Evaluation Criteria:**

Yes, the proposed methods and evaluation criteria presented in the Falcon paper generally make sense for the stated problem of accelerating diffusion-based visuomotor policies for robotic tasks.

The authors evaluate Falcon on diverse and representative benchmark datasets, each suitable for validating different aspects of visuomotor policy performance.

While the current evaluation criteria are robust and appropriate, the authors could further strengthen the evaluation: Include at least limited real-robot experiments to demonstrate practical utility, robustness, and generalization beyond simulated environments.

Maintaining a buffer of previously denoised actions could significantly increase memory consumption. This may limit practical deployment in resource-constrained scenarios.

**Other Comments Or Suggestions:**

Figure 1 is somewhat complex and could benefit from simplification or clearer annotation.

**Other Strengths And Weaknesses:**

Falcon creatively combines existing ideas from diffusion models, Tweedie's formula, and sequential dependency exploitation. Although each component individually is known, the particular combination is novel and cleverly addresses a key practical challenge—slow inference speed.

However, the paper does not sufficiently address the complexity, memory overhead, and scalability concerns introduced by buffer management. Furthermore, the approach lacks validation through real-world robotic tasks, as evaluations remain limited to simulated environments.

**Questions For Authors:**

Falcon currently requires manual tuning of hyperparameters. Could you clarify how sensitive Falcon's performance is to these hyperparameters across diverse tasks, and discuss whether you've explored any automated or adaptive methods for tuning them?

Could you clarify the computational and memory overhead of maintaining the latent buffer in practical deployment scenarios?

**Relation To Broader Scientific Literature:**

The contributions of the Falcon paper are closely related to several areas of existing research in diffusion models and robotic policy learning.

Falcon builds explicitly upon the foundational formulations of Denoising Diffusion Probabilistic Models (DDPM, Ho et al., 2020) and Denoising Diffusion Implicit Models (DDIM, Song et al., 2020). Existing methods such as DDIM (Song et al., 2020a) and DPMSolver (Lu et al., 2022) accelerate diffusion sampling by reducing the number of iterative denoising steps. Falcon explicitly integrates with and further enhances these existing solvers.

Diffusion policies (Chi et al., 2023) inherently model multimodal action distributions. Falcon retains this strength through careful initialization and partial denoising, contrasting with distillation methods that often reduce multimodality.

**Theoretical Claims:**

The derivations and mathematical transformations in the paper appear to be correct, and consistent with existing literature on diffusion-based methods.

---

> ### Author Rebuttal · Authors · 2025-03-31
>
> We sincerely thank the reviewer for the detailed and thoughtful feedback. We appreciate your recognition of Falcon’s novelty and practical motivation, and we are grateful for your constructive suggestions on deployment, which helped us strengthen the final version of our work.
>
> **Real-World Validation**
>
> To address the concern about practical utility, we conducted a **real-world dexterous grasping** using a physical robotic setup (see in **Fig. 1 of the [link](https://anonymous.4open.science/api/repo/506Falcon/file/material.pdf?v=69217ede)**) . Falcon, DDPM, and SDP were evaluated using the same Unet backbone with planning horizon $T_p=16$. As shown in **Table 1 on our [link](https://anonymous.4open.science/api/repo/506Falcon/file/material.pdf?v=69217ede)**, all methods achieved 100% success, while **Falcon reduced runtime by 3.07x compared to DDPM**, and **outperformed SDP in both runtime (0.14s vs. 0.23s) and memory usage (3730MB vs. 3808MB)**, demonstrating strong real-world performance.
>
> **Memory and Computational Overhead**
>
> Falcon maintains a **small buffer of 20–50 actions** (**see Appendix D in our submission**), while SDP requires a much larger horizon × noise-level buffer (e.g., 16×100). As shown in **Table 7 of the [link](https://anonymous.4open.science/api/repo/506Falcon/file/material.pdf?v=69217ede)**, Falcon’s memory overhead is **+12MB**, significantly lower than SDP’s **+26MB**, making Falcon more suitable for real-time and resource-constrained deployment.
>
> **Hyperparameter Sensitivity and Usability**
>
> Falcon introduces two scalar hyperparameters: $\epsilon$ (reuse threshold) and $\delta$ (exploration rate), which control the balance between inference speed and action accuracy.. While both influence the trade-off between speed and accuracy, our ablations (**Figure 3, left and middle in submission**) show that **Falcon performs robustly when ε ∈ [0.001, 0.01] and δ ∈ [0.1, 0.2]**, across all tested environments. These parameters are intuitive, require **no retraining or fine-tuning**, and generalize well across tasks. We will include these practical guidelines in the revision.
>
> **Comparison with Streaming Diffusion Policy (SDP)**
>
> We thank the reviewer for highlighting SDP [1], which is indeed an important related work. We would like to clarify that SDP was already **cited and briefly discussed in Section 5 of our submission**. We appreciate the opportunity to strengthen this comparison further.
>
> While both Falcon and SDP leverage partial denoising for policy acceleration, Falcon offers **2 core advantages:**
>
> - **Training-free and plug-and-play**:
>     Falcon does **not require task-specific training** or modifications to the diffusion model. In contrast, **SDP must be retrained** on each task with a specially designed noise corruption scheme (see SDP Section 3.3) to enable recursive denoising. This limits SDP’s adaptability and ease of deployment.
> - **Lower memory overhead**:
>     Falcon uses a **threshold-based selection mechanism** to maintain a compact buffer (20–50 entries). SDP, by design, stores a full horizon × noise-level buffer (e.g., 16×100 entries). As shown in **Table 7 on our [link](https://anonymous.4open.science/api/repo/506Falcon/file/material.pdf?v=69217ede)**, Falcon adds only **+12MB** of memory compared to DDPM, while SDP adds **+26MB**—a significant difference for resource-constrained settings.
>
> In both **simulated tasks** (**Tables 4–8 in our [link](https://anonymous.4open.science/api/repo/506Falcon/file/material.pdf?v=69217ede)**) and **real-world experiments** (**Table 1 in our [link](https://anonymous.4open.science/api/repo/506Falcon/file/material.pdf?v=69217ede)**), Falcon achieves **comparable or higher success rates than SDP**, while also achieving **higher speedups and lower runtime** (e.g., 0.14s vs. 0.23s in real-world grasping, . These results demonstrate that Falcon not only accelerates diffusion policies but also transfers effectively to physical robot systems.
>
> **Figure Improvement (Regarding Figure 1)**
>
> Thank you for the helpful suggestion. We have revised **Figure 1** to improve clarity and visual structure, now included in our updated submission and in the **[link](https://anonymous.4open.science/api/repo/506Falcon/file/material.pdf?v=69217ede) (Figure 3)**.
>
> The new diagram adopts a cleaner modular layout and color-coded elements to distinguish the two-stage mechanism: (1) reference action estimation and (2) threshold-based candidate selection. We believe this revision significantly improves readability.
>
> **Closing Remarks**
>
> We once again thank the reviewer for their valuable feedback. If our responses have addressed your concerns, we would be sincerely grateful for your consideration in revising the score. Please let us know if further clarification would be helpful—we would be happy to provide it.
>
> **References**
>
> [1] Fast Policy Synthesis with Variable Noise Diffusion Models

---

> > ### Comment · Reviewer_8uiC · 2025-04-03
> >
> > The authors have addressed most of my questions. I will keep my current rating for this paper.

---

> > > ### Author Response · Authors · 2025-04-09
> > >
> > > We appreciate your efforts in reviewing our paper and rebuttal, and thank you for your valuable feedback!

---

### Official Review · Reviewer_7PAa · 2025-03-10

**Overall Recommendation:** 4

**Summary:**

This paper presents Falcon, Fast visuomotor policies via partial denoising. This approach improves diffusion policies by accelerating action generation while preserving the multimodal generation capability. Accelerations are mainly provided by using partial denoised actions to reduce denoising steps. Falcon is a training-free algorithm and can be plugged-in to further improve efficiency on top of existing techniques. The proposed algorithm has been evaluated on three different simulated robotics datasets that expose different challenges and help evaluate the contributions of this paper.


## update after rebuttal
I confirm my score. Authors addressed comments and added clarity and results to the original submission.

**Claims And Evidence:**

Claims are supported by clear and convincing evidence, through thorough analysis and grounding the proposed approach in relevant literature.

**Essential References Not Discussed:**

N/A

**Experimental Designs Or Analyses:**

Experiments are meaningful and informative. Results are analyzed in a compelling way and ablations are useful and to the point.

**Methods And Evaluation Criteria:**

The proposed method is supported by clear and convincing evaluations, through experiments on three robotics dataset, ablations of parameters, analysis of results presented in a clear way.
Note: the two parameters (epsilon and delta) play a key role in obtaining speed and high success score. Is there an analysis of their relationship or an hypothesis regarding how to set guarantees ranges for these two in order to achieve high scores and speed?

**Other Comments Or Suggestions:**

See note on definition of alpha.

**Other Strengths And Weaknesses:**

The paper is well presented and significant. Claims, theoretical contributions and experiments are presented in a clear way. It can have impact on robotics applications.

**Questions For Authors:**

Please refer to the other comments.

**Relation To Broader Scientific Literature:**

This work is relevant for the robotics community where diffusion policies can play an important role.

**Theoretical Claims:**

I reviewed the theoretical claims and equations, although I did not verify all mathematical derivations in detail. While it is possible that I may have overlooked some aspects, the theoretical claims appear to be correct to the best of my understanding.
Note: alpha (page 2) is not defined - please add a definition or relevant reference.

---

> ### Author Rebuttal · Authors · 2025-03-31
>
> We sincerely thank the reviewer for the thoughtful suggestion and for the generous score. We deeply appreciate your recognition of our work, and your feedback has helped us identify opportunities to clarify and strengthen our contributions.
>
> **On the Relationship Between $\epsilon$ and $\delta$**
>
> In Falcon, $\epsilon$ and $\delta$ play complementary roles in controlling the reuse of partial denoised actions.
>
> - **$\epsilon$ serves as a selection threshold** that determines whether a previously denoised action is temporally aligned enough to be reused. Smaller ε enforces stricter matching (favoring accuracy), while larger $\epsilon$ allows more aggressive reuse (favoring speed).
>
> - **$\delta$ is the exploration rate**, specifying the probability of discarding the reused action and instead sampling from standard Gaussian noise. This mechanism injects diversity into the action sequence and prevents over-reliance on suboptimal reuse.
>
> While our ablations (**Figure 3, left and middle in our submission**) analyze these parameters independently, we agree that their joint effect is important. In our experiments, we typically fix **$\delta = 0.1$ or $\delta = 0.2$** , and find that **$\epsilon \in [0.001, 0.01]$** works robustly across tasks. This setting consistently balances acceleration and success rate. We will explore their interaction more formally in future work.
>
> **On the Definition of $\alpha$**
>
> Thank you also for pointing out the missing definition of $\alpha$. It refers to the variance schedule in the forward diffusion process, as commonly defined in DDPM[1] or DDIM[2].   We will include the formal definition and cite the appropriate reference in the revised version.
>
> We once again thank the reviewer for their thoughtful and constructive feedback. Your comments have been instrumental in helping us refine the clarity and usability of our method.
>
>
>
> **References**
>
> [1] Denoising Diffusion Probabilistic Models
>
> [2] Denoising Diffusion Implicit Models

---

> > ### Comment · Reviewer_7PAa · 2025-04-02
> >
> > Thank you for addressing my and other reviewers' comments.

---

> > > ### Author Response · Authors · 2025-04-09
> > >
> > > Thank you for your time and effort! We really appreciate your support in our work!

---

### Official Review · Reviewer_4p76 · 2025-03-13

**Overall Recommendation:** 3

**Summary:**

This paper introduces Falcon, a method that accelerates the diffusion process by denoising partially noisy actions at each step using a one-step adaptive mechanism. Extensive experiments validate the speed improvement of the Falcon method on robot datasets.

## Update after rebuttal
I'm satisfied with the authors' rebuttal which addresses most of my main concerns, including the comparisons with related baselines and the real-world experiments.

**Claims And Evidence:**

Yes.

**Essential References Not Discussed:**

Streaming Diffusion Policy (ICRA 2025, https://arxiv.org/pdf/2503.04051), which also leveraged the insight that generating a partially denoised action trajectory is substantially faster than a full output action trajectory, and was released last year.

**Experimental Designs Or Analyses:**

The main experiments in the paper are conducted on the RoboMimic dataset, with comparatively fewer analyses on other datasets. I would have liked to see Falcon's performance on more complex datasets, such as Dexterous Grasping or Long-Horizon tasks.

**Methods And Evaluation Criteria:**

Yes.

**Other Comments Or Suggestions:**

Overall, this paper presents an interesting idea of using partial noise to accelerate diffusion policies. However, the emergence of existing work (SDP) and the performance drop caused by the filtering mechanism reduce the novelty of this work. Nevertheless, I still look forward to the authors providing a more fundamental explanation of why Falcon causes uncontrollable performance degradation and potential solutions for hyperparameter sensitivity. Additionally, I would like to see more real-world experiments. I will adjust my future rating based on the improvements made.

If I have misunderstood any points, I am open to discussion.

**Other Strengths And Weaknesses:**

**Strengths:**

- The paper is well-written and easy to read.
- The simulation experiments are sufficient and thorough.
- The motivation is clear and reasonable, aiming to accelerate the diffusion denoising process using partially noisy actions. The corresponding mathematical tools support the proposed method.

**Weaknesses:**

- One of the major issues with this paper is that despite having a strict selection mechanism (Line 12 in Algorithm 1), the accuracy drop caused by the partial noise selected through this mechanism is significant. Specifically, in Table 1 (DDIM + Falcon), this not only raises questions about the effectiveness of the threshold mechanism but also calls for a more detailed explanation from the authors about why this limitation still results in performance degradation with carefully selected partial noise candidates.

- This paper needs a more detailed comparison with existing work on streaming diffusion policies (SDP [1]), which uses step-wise partial noise without filtering as a buffer to achieve acceleration. A basic partial noise arrangement, such as the one in SDP where the noise level decreases as $t$ increases, should serve as a baseline to validate the effectiveness of the threshold mechanism.

- There is a lack of more comparative results on the hyperparameter epsilon. The existing results indicate that the method is highly sensitive to this hyperparameter, which poses challenges for deployment on different tasks in the future.

- As a robot learning paper, it is difficult to be highly convinced of the method's real-world effectiveness without physical experiments.

- Please consider comparing with previous partial denoising-based methods [1][2]. If some of the works are considered concurrent works, feel free to disregard them.

---
*References:*

[1] Fast Policy Synthesis with Variable Noise Diffusion Models. https://arxiv.org/pdf/2406.04806

[2] Responsive Noise-Relaying Diffusion Policy: Responsive and Efficient Visuomotor Control. https://arxiv.org/pdf/2502.12724

**Questions For Authors:**

Please carefully read the weaknesses part.

**Relation To Broader Scientific Literature:**

Acceleration of diffusion policies is crucial in the field of robotics, as lower latency benefits practical deployment in real-world robotic systems.

**Theoretical Claims:**

The provided propositions are clear and have been proven.

---

> ### Author Rebuttal · Authors · 2025-03-31
>
> We sincerely thank the reviewer for the thoughtful and constructive feedback. We appreciate your recognition of Falcon’s motivation and your suggestions for strengthening the comparison and evaluation. Below, we address each concern in turn.
>
> **On the Effectiveness of the Threshold Mechanism (DDIM+Falcon)**
>
> The performance drop in **Table 1** for DDIM+Falcon stems from the threshold $\epsilon $ being set relatively high in our original configuration to maximize speedup. This allowed partial denoised actions with weaker temporal dependency to be reused, leading to performance degradation. This does not indicate a flaw in the mechanism itself, but rather reflects a trade-off between efficiency and accuracy.
>
> To address this, we conducted additional experiments with smaller $\epsilon$ values. As shown in **Table 9 in the [link](https://anonymous.4open.science/api/repo/506Falcon/file/material.pdf?v=69217ede)**, reducing $\epsilon$  (e.g., **0.005 → 0.001, 0.01 → 0.003**) significantly improved success rates in Transport **(0.74 → 0.81)** and Tool Hang **(0.51 → 0.54)**, while retaining acceleration.
>
> We also note that **Figure 3 (left)** in our main paper presents an ablation study showing this trade-off trend. For more task-specific results, **Figure 5 in the [link](https://anonymous.4open.science/api/repo/506Falcon/file/material.pdf?v=69217ede)** further supports this behavior. We will incorporate the updated results in the revised version.
>
> **Comparison with SDP[1]**
>
> We would like to clarify that **SDP was already cited and briefly discussed in Section 5 of the original submission**. We appreciate the opportunity to provide a more detailed comparison.
>
> Falcon offers **two key advantages** over SDP:
>
> - **Training-free**
>
>     Falcon requires no retraining or noise-corruption design. In contrast, **SDP must be retrained per task** with a handcrafted noise corruption scheme (SDP Sec. 3.3), limiting flexibility.
>
> - **Less Memory Cost**
>
>     **Falcon’s thresholding mechanism allows it to maintain a small buffer (e.g. 50, see Appendix D)**, while SDP stores a full horizon × noise-level buffer (e.g., 16×100). As shown in **Table 7 on our [link](https://anonymous.4open.science/api/repo/506Falcon/file/material.pdf?v=69217ede)**, Falcon adds only +12MB of memory compared with DDPM, but SDP adds + 26M.
>
> Finally, we implemented SDP under our setting and provide comparisons in **Tables 4–8 on our [link](https://anonymous.4open.science/api/repo/506Falcon/file/material.pdf?v=69217ede)**. Falcon achieves comparable task performance while offering higher speedups and lower memory overhead.
>
> **On Sensitivity to $\epsilon$ and Deployment Concerns**
>
> We agree that $\epsilon$ is a key hyperparameter, and we have already provided a detailed analysis of its impact in **Figure 3 (left)** in our submission and **Figure 5 in [link](https://anonymous.4open.science/api/repo/506Falcon/file/material.pdf?v=69217ede)**. These results show that:
>
> - Falcon’s performance remains stable within a **broad $\epsilon$ range (e.g., 0.001–0.01)** across tasks;
> - The accuracy-speed trade-off is smooth and interpretable, making ε easy to configure in practice;
> - The same $\epsilon$ values generalize well across different environments, reducing per-task tuning burden.
>
> Although Falcon does rely on $\epsilon$ , it is **training-free**, and introduces only a few parameter to adjust—a significantly lighter requirement compared to training based acceleration methods.
>
> We believe this controllable trade-off offers a practical balance between deployment flexibility and performance.
>
> **On Real-World Deployment Evaluation**
>
> To assess real-world applicability, we conducted a **dexterous grasping experiment** with a physical robotic platform (**Figure 1 in [link](https://anonymous.4open.science/api/repo/506Falcon/file/material.pdf?v=69217ede)**), composed of a RealMan 7-DoF arm, a PsiBot 6-DoF hand, and dual RealSense cameras.
>
> DDPM, SDP(DDPM), and Falcon+DDPM were deployed using the same Unet architecture, with $T_p=16,T_o=1,T_a=8$. As shown in **Table 1 on our [link](https://anonymous.4open.science/api/repo/506Falcon/file/material.pdf?v=69217ede)**, all methods achieved 100% success rate, but **Falcon achieved 3.07× speedup** and outperformed SDP in both runtime **(0.14s vs. 0.23s)** and memory consumption **(3730MB vs. 3808MB CPU)**.
>
> These results confirm that Falcon not only accelerates diffusion policies in simulation but also transfers efficiently to real-world robotic execution.
>
> **Others**
>
> We consider Paper [2] as concurrent works.
>
> **Closing Remarks**
>
> We once again thank the reviewer for the detailed comments. If our responses have addressed your concerns, we would be sincerely grateful for your consideration in revising the score. Please let us know if further clarification is needed.
>
> **References:**
>
> [1] Fast Policy Synthesis with Variable Noise Diffusion Models
>
> [2] Responsive Noise-Relaying Diffusion Policy

---

> > ### Comment · Reviewer_4p76 · 2025-04-02
> >
> > Dear authors,
> >
> > I have read the comments of each reviewer and checked the rebuttal file very carefully. I am truly impressed and grateful that the authors were able to address all concerns—including those regarding real-world experiments—in such a short timeframe. I understand how much effort and dedication this must have required, as preparing such comprehensive responses under tight deadlines is both mentally taxing and time-consuming. The authors have fully resolved my concerns, and I will increase my score without hesitation.
> >
> > Although the rebuttal satisfactorily addresses the current issues, I would like to highlight a few points for further discussion that could help refine the final version of this paper and enhance its professionalism.
> >
> > **Suggestions about Threshold Mechanism:**
> >
> > As I expected, the proposed method demonstrates stronger capabilities than the baselines after the ε is carefully selected, which is reasonable given its carefully designed adaptive mechanism. The results show that smaller values of ε (~0.00X) consistently yield better performance. However, this raises an important question (which I also mentioned in Weakness 3): the selection of ε remains somewhat difficult to control precisely. The results suggest that an optimal ε must be determined for each individual task, which would require extensive manual pre-testing if the number of downstream tasks is large.
> >
> > To address this, I offer the following suggestions:
> >
> > -  Investigate whether a more universal ε can be derived—one that accommodates all candidate actions while maintaining sufficient performance, from a theoretical perspective. For instance, integrating the probability density function (i.e., the area) uniformly could ensure that the gap between the actual action and the reference action remains within an acceptable range.
> >
> > -  Explore the development of a more sophisticated ε-selection algorithm, such as one that adapts dynamically based on the observation or robot state input.
> >
> > - (A small suggestion for future work) Extend this mechanism to more generalized diffusion policies (e.g., RDT or π-zero, though the latter is based on flow matching) to eliminate the need for per-task ε selection.
> >
> > **Suggestions about Real-world Experiments:**
> >
> > I sincerely appreciate the authors’ efforts in conducting real-world experiments within such a constrained timeframe. However, I noticed one additional issue in the rebuttal: the reported success rates for all methods are 100%. This may give the impression that the experimental tasks are overly simplistic, potentially obscuring the algorithm’s ability to differentiate itself from baselines. Additionally, the real-world results show only marginal differences in acceleration (including speed and memory usage) between SDP and Falcon, which might be negligible in practice.
> >
> > To strengthen the paper, I suggest incorporating more complex real-world tasks in future versions. This would better demonstrate Falcon’s superior performance not only in accuracy but also in computational efficiency.
> >
> > ---
> >
> > Once again, I commend the authors for their diligent work and thoughtful revisions. I am confident that addressing these remaining points will further elevate the impact and clarity of this already impressive research. I look forward to seeing the final version and the continued advancements in this direction.
> >
> > Best,
> >
> > Reviewer 4p76

---

> > > ### Author Response · Authors · 2025-04-09
> > >
> > > We sincerely thank the reviewer for the encouraging follow-up and the generous score increase. To ensure clarity, we have uploaded **new results and figures** to our supplementary site: https://anonymous.4open.science/api/repo/506Falcon/file/material2.pdf?v=6a54d6d0
> > >
> > > Below, we respond to your suggestions:
> > >
> > > **Generalization to RDT (Foundation Diffusion Policy)**
> > >
> > > To demonstrate Falcon’s compatibility with task-generalizable diffusion models, we applied it to RDT[3] (a pretrained vla model), integrating Falcon into RDT’s DPMSolver++ without retraining. As shown in **Table 10, Fig. 6  and Fig. 7of [link2](https://anonymous.4open.science/api/repo/506Falcon/file/material2.pdf?v=6a54d6d0)**, Falcon significantly accelerates inference (**31x and 34x vs. DDPM**) **on PickCube and PushCube tasks** while preserving success rate, showing strong generalization.
> > >
> > > ------
> > >
> > > | Method                         | Task     | Success Rate | NFE  | Time(s) | Speedup | GPU MEMS (MB) |
> > > | ------------------------------ | -------- | ------------ | ---- | ------- | ------- | ------------- |
> > > | DPMsolver++ (5 steps)          | PickCube | 0.75         | 5.00 | 0.08    | 20×     | 15479         |
> > > | DPMsolver++ (5 steps) + Falcon | PickCube | 0.75         | 3.22 | 0.04    | 31x     | 15483         |
> > > | DPMsolver++ (5 steps)          | PushCube | 1.00         | 5.00 | 0.08    | 20×     | 15479         |
> > > | DPMsolver++ (5 steps) + Falcon | PushCube | 1.00         | 2.91 | 0.05    | 34x     | 15483         |
> > >
> > > **Table 10**: Falcon is evaluated with 20 rollouts on ManiSkill benchmark ($T_p=64, T_a=32, T_o=2, \epsilon=0.02,|\mathcal{B}|=2$). Speedup is relative to 100-step DDPM.
> > >
> > > ------
> > >
> > >
> > >
> > > **More Challenging Real-World Evaluation**
> > >
> > > Building on your suggestion, we conducted an additional complex real-world experiment involving **precise object insertion (see in Fig. 1 of [link2](https://anonymous.4open.science/api/repo/506Falcon/file/material2.pdf?v=6a54d6d0))**. The robot must insert a **square stick** into a tall **chip can**—a task requiring accurate 3D alignment. Even minor errors in angle or height result in failure.
> > >
> > > We trained DDPM and SDP on 50 human demonstrations, and applied Falcon on top of the trained DDPM (no retraining needed). Falcon matched DDPM in success rate (90%) but with **2.86× faster inference** (see **Table 11** and **Fig. 8** in the **[link2](https://anonymous.4open.science/api/repo/506Falcon/file/material2.pdf?v=6a54d6d0)**). Falcon maintained 90% success rate while being **2.86× faster**.
> > >
> > > ------
> > >
> > > | Method          | Speedup   | NFE              | Sampling Time per action (s) | GPU MEMS (MB)  | Success rate |
> > > | --------------- | --------- | ---------------- | ---------------------------- | -------------- | ------------ |
> > > | DDPM            | 1.00×     | 100.00 ± 0.00    | 0.43 ± 0.01                  | 3735.76 ± 0.48 | 90%          |
> > > | SDP(DDPM)       | 1.95×     | 50.00 ± 0.00     | 0.22 ± 0.01                  | 3743.28 ± 1.26 | 85%          |
> > > | **Falcon+DDPM** | **2.86×** | **25.57 ± 7.10** | **0.15 ± 0.12**              | 3731.50 ± 5.08 | 90%          |
> > >
> > > **Table 11**: Each entry is evaluated with 20 rollouts in the mean $\pm$ standard deviation format. Falcon is set with $T_p=32,T_a=16,T_o=1,\epsilon=0.02, |\mathcal{B}|=20$.
> > >
> > > ------
> > >
> > >
> > >
> > > **On $\epsilon$ Selection Strategy**
> > >
> > > We deeply appreciate your suggestions regarding the difficulty of manually tuning $\epsilon$. In practice, we observed that there exists value within $\epsilon \in [0.001, 0.05]$  yield good performance across tasks (see in **Fig. 5 in [link2](https://anonymous.4open.science/api/repo/506Falcon/file/material2.pdf?v=6a54d6d0) , Fig. 3 in submission**, with a predictable trade-off: **larger $\epsilon$ increases speedup**, while **smaller $\epsilon$ favors accuracy**.
> > >
> > > To ease this tuning process, we recommend a **binary search strategy**—given the monotonic behavior of performance with respect to ε, this approach has a **logarithmic time complexity** and requires only a small number of trials. Since Falcon is **training-free**, this tuning is light-weight and incurs minimal cost in practice.
> > >
> > > We fully agree with the importance of automating $\epsilon$ selection, and your suggestion has inspired us to pursue more **adaptive strategies** that leverage observation statistics or score distributions during inference. Due to the limited rebuttal timeframe, we were unable to fully implement and test these ideas, but we are actively exploring this direction as part of our future work and will continue extending Falcon to more complex real-world tasks.
> > >
> > > We are grateful for your thoughtful feedback, which helped us better articulate and extend the applicability of Falcon. If our updates have resolved your remaining concerns, we would greatly appreciate your further support and score reconsideration.
> > >
> > > **References**
> > >
> > > [3] RDT-1B: a diffusion foundation model for bimanual manipulation

---

### Official Review · Reviewer_i58m · 2025-03-14

**Overall Recommendation:** 2

**Summary:**

The authors propose a method to speed up inference with diffusion policies using a scheme where the denoising chain for action sequences is initialized to a partially noised sequence predicted from a previous timestep. The proposed method is supposed to preserve the multimodality of the diffusion policy while achieving anywhere from 2 to 7x speedup. The authors evaluate their method on a variety of simulated robotics tasks.

**Claims And Evidence:**

The claims regarding significant speedup are clear. However, ultimately I do not find the proposed method convincing. The authors motivate their method by claiming that other more principled approaches to speed up diffusion inference such as ODE solvers suffer from numerical discretisation errors, and distillation techniques cannot represent multimodal policy distribution. These claims are never properly justified in the paper. The proposed method itself is does not have much theoretical motivation either.

**Essential References Not Discussed:**

The literature in fast diffusion sampling is vast, and for flow models there are even more. The authors should additionally cite progressive distillation at least, since it is a very popular and important paper in the field.

**Experimental Designs Or Analyses:**

As mentioned earlier, the tasks themselves are probably fine, as long as a Gaussian baseline is included to show that they actually have multimodal behavior policies. Table 2 compares number of function evaluations, and for DPMSolver and DDPM more evaluations are shown to be used then Falcon. However, we can see that NFE for DPM solver is already quite close to the Falcon. No comparison is shown where the ODE in DPMSolver is integrated with fewer steps. Does it actually lead to loss in performance in these tasks? There is also no distillation method that is compared against (consistency distillation, progressive distillation etc.) except a very limited evaluation in Fig 4, which in theory preserve the same marginals as the original diffusion model so should see minimal performance drop.

**Methods And Evaluation Criteria:**

The datasets and tasks which are evaluated are a good starting point, but there are some issues. Firstly, there is no evidence provided that the behavior policy for these datasets is actually multimodal, and requires diffusion policies. Section 4.5 does not truly evaluate multimodality of policy distributions, since trajectory level multimodality can even be achieved through per-action unimodal policies like Gaussians. It is important to have a Gaussian policy baseline to compare against. One of the important claims of the paper is that the proposed method preserves multimodality whereas other distillation methods do not, and this claim is never backed up.

**Other Comments Or Suggestions:**

Minor error: in line 432 the consistency policy reference instead points to consistency trajectory models, which is a different paper from the consistency policy paper (which is referenced correctly in section 4.5).

**Other Strengths And Weaknesses:**

### Strengths
1. The authors evaluate on many different tasks, which is appreciated

### Weaknesses
1. The proposed method is explained poorly in my opinion. I think this has more to do with the method not having sufficient theoretical justification.
2. Some confusing statements which are not really justified, such as "Moreover, distillation-based approaches are inherently training-intensive and task-specific, meaning they cannot generalize effectively to accelerate unseen tasks or adapt to diverse visuomotor applications."  in the introduction.

**Questions For Authors:**

1. Why do you say distillation techniques are task specific and not generally applicable? Most diffusion distillation techniques are applicable to any diffusion model.
2. How do you know your proposed method requires less function evaluations than DPMSolver? You can manually adjust the number of steps for the ODE solvers using different integration schemes.

**Relation To Broader Scientific Literature:**

The paper is related to fast sampling of diffusion models. Some important works in this class are ODE based sampling [1], ODE distillation [2, 3], and progressive distillation [4] among many others.

[1] DPM-Solver: A Fast ODE Solver for Diffusion Probabilistic Model Sampling in Around 10 Steps, https://arxiv.org/abs/2206.00927

[2] Consistency Models as a Rich and Efficient Policy Class for Reinforcement Learning, https://arxiv.org/abs/2309.16984

[3] Consistency Trajectory Models: Learning Probability Flow ODE Trajectory of Diffusion, https://arxiv.org/abs/2310.02279

[4] Progressive Distillation for Fast Sampling of Diffusion Models, https://arxiv.org/abs/2202.00512

**Theoretical Claims:**

There are no theoretical claims made by the paper. The proposed method is justified primarily through author's intuition and empirical evaluation.

---

> ### Author Rebuttal · Authors · 2025-03-31
>
> We sincerely thank the reviewer for the thoughtful and detailed comments. We appreciate the constructive feedback and address the key concern below.
>
> **Clarifying the Motivation and Positioning of Our Method**
>
> Our contribution is not to replace distillation or ODE solvers, but to offer a practical, training-free alternative that leverages temporal structure for faster inference in diffusion policies.
>
> While we discuss limitations of prior methods, Falcon is not motivated by them. As stated in our abstract, *“The core insight is that visuomotor tasks exhibit sequential dependencies between actions at consecutive time steps.”* Falcon leverages this via partial denoising to reduce sampling steps.
>
> We fully recognize that **distillation methods are powerful**, particularly in settings with large data and compute [2,4,5,6]. In such settings, they deliver substantial speedups while maintaining performance. We respect this line of work and do not position Falcon as a replacement.
>
> Instead, Falcon instead targets low-resource regimes. In these cases, distillation may be infeasible, and ODE solvers can suffer from large discretization errors under very low NFE [1]. Falcon addresses this gap by offering a plug-and-play solution that also integrates well with ODE solvers in low-step regimes **(Section 4.3)**.
>
> We acknowledge that our initial wording may have been overly strong regarding distillation. While methods like CP [2, **Sec. 5**] may face challenges in preserving multimodality, this is task-dependent. We will revise accordingly, and thank the reviewer for pointing this out.
>
> **Lack of Theoretical Foundation**
>
> Although Falcon lacks a full theoretical derivation, its **core components are grounded in well-established principles**.
>
> Falcon is built on the insight that in sequential decision-making, when past actions are strongly correlated with the current one, initializing from them can reduce sampling steps. Falcon exploits this via reusing past denoised actions.
>
> Crucially, the choice of which past action to reuse is not heuristic. It is determined by Falcon’s **thresholding mechanism**, based on **Tweedie’s formula**, a foundational result in empirical Bayes theory. As noted in **Remark 1 of [7]**, Tweedie’s formula yields the Bayes-optimal posterior mean under Gaussian noise, guiding our selection of the cleanest available prior. This forms the theoretical core of our partial denoising module.
>
> Thus, while Falcon lacks a full pipeline of theory, its **main mechanism—Tweedie-based partial action reuse—is mathematically justified** and validated through strong empirical performance.
>
> **On Expressing Multimodal Distributions**
>
> 1. The BlockPushing task has inherently multimodal expert trajectories (**BeT [3], Table 2**).
> 2. We added a **Gaussian baseline** using the same network. It fails on both goals **(p1: 0.02, p2: 0.01)**, while Falcon succeeds **(p1: 0.99, p2: 0.97), (see in Table 3 in the [link](https://anonymous.4open.science/api/repo/506Falcon/file/material.pdf?v=69217ede))** showing the need for multimodal modeling.
> 3. We evaluated **Consistency Policy** on PushT and found it biased toward one mode, indicating that some distillation methods may struggle to retain multimodality see in **Figure 4 in the [link](https://anonymous.4open.science/api/repo/506Falcon/file/material.pdf?v=69217ede)**.
>
> **Comparisons with Reduced-Step DPMSolver**
>
> To directly assess whether lower-step DPMSolver leads to performance degradation, we matched its NFE to that of DPMSolver+Falcon. Under this constraint, DPM-Solver* exhibited significant performance drops—**10.6%** in Square_ph, **35.6%** in Square_mh, and **6.8%** in Transport_ph—while Falcon maintained high success rates (**see Table 2 in [link](https://anonymous.4open.science/api/repo/506Falcon/file/material.pdf?v=69217ede)**). This confirms that aggressive NFE reduction in ODE solvers can hurt performance, and Falcon helps mitigate this degradation.
>
> **Missing or Incorrect References**
>
> We will add the missing citation for Progressive Distillation [6], and correct the mistaken reference on Line 432.
>
> **Closing Remarks**
>
> We are grateful for the reviewer’s thoughtful comments, which helped us improve both the clarity and positioning of our work. If our responses have addressed your concerns, we would greatly appreciate a reconsideration of the score. Please let us know if further clarification is needed.
>
> **References**
>
> [1] PFDiff: Training-Free Acceleration of Diffusion Models Combining Past and Future Scores
>
> [2] Consistency Policy: Accelerated Visuomotor Policies via Consistency Distillation
>
> [3] Behavior Transformers: Cloning k Modes with One Stone
>
> [4] Consistency Models as a Rich and Efficient Policy Class for Reinforcement Learning
>
> [5] Consistency Trajectory Models: Learning Probability Flow ODE Trajectory of Diffusion
>
> [6] Progressive Distillation for Fast Sampling of Diffusion Models
>
> [7] Diffusion Posterior Sampling for General Noisy Inverse Problems

---

> > ### Comment · Reviewer_i58m · 2025-04-04
> >
> > I thank the authors for the response, however many of my original problems with the paper remain. I fundamentally do not think the contribution is significant, however I can raise the score to weak reject.

---

> > > ### Author Response · Authors · 2025-04-09
> > >
> > > We sincerely thank you for the follow-up response.
> > >
> > > The concerns you raised—including **comparisons with distillation methods**, **the expressiveness of Gaussian baselines**, and **lower-step DPMSolver**—have been addressed through new experiments and revisions. Regarding the concern that our method is not convincing, we have conducted extensive evaluations including: (1) **comparisons with SDP[1] and Consistency Policy[2]**, (2) **Gaussian baseline and multimodal trajectory analysis**, (3) **two real-world robot experiments (dexterous grasping and insertion)**, and (4) **a vision-lanugage-action foundation model RDT[3] experiment**. Several of these were also requested by other reviewers, and we are glad to report that they are now included in the updated version.
> > >
> > > Once again, we truly appreciate your feedback—and if our updates have addressed your concerns, we would be sincerely grateful for your reconsideration of the score.
> > >
> > > ------
> > >
> > > All referenced experiments are available in our supplementary material: https://anonymous.4open.science/api/repo/506Falcon/file/material2.pdf?v=6a54d6d0:
> > >
> > > **Distillation comparison**: see **Fig. 4**
> > >
> > > **Gaussian baseline**: see **Table 3**
> > >
> > > **Lower-step DPMSolver**: see **Table 2**
> > >
> > > **SDP comparisons**: see **Tables 4–7**
> > >
> > > **Two real-world robot experiments**: see **Figs. 1, 2, 8** and **Tables 1, 11**
> > >
> > > **Vision-language-action foundation model experiment (RDT)**: see **Figs. 6, 7** and **Table 10**
> > >
> > > ------
> > >
> > > References:
> > >
> > > [1] Fast Policy Synthesis with Variable Noise Diffusion Models
> > >
> > > [2] Consistency Policy: Accelerated Visuomotor Policies via Consistency Distillation
> > >
> > > [3] RDT-1B: a diffusion foundation model for bimanual manipulation

---

### Decision · Program_Chairs · 2025-05-01

**Decision:**

Accept (poster)

**Comment:**

This paper proposes a method to speed up inference with diffusion policies while preserving multimodality.
Several gaps in the empirical evaluation emerged during the review phase, but the authors have thoroughly addressed them. I'm positive that adding the additional results to the final version will make it much stronger.

There remains an issue with the motivation of the proposed approach: claims on the limitations of existing acceleration techniques are not fully and convincingly supported by empirical evidence. For the final version, some further work is needed to adjust the narrative of the paper and clarify the intent and scope of the contribution, as the authors wrote "not to replace distillation or ODE solvers, but to offer a practical, training-free alternative that leverages temporal structure for faster inference in diffusion policies".
Given the effort already put by the authors in the rebuttal phase, I trust them on fixing this presentation issue for the camera-ready. In doing so, please also consider that some observed shortcomings of alternative methods (e.g. mode collapse of consistency policies) may be due to the implementation. I suggest to acknowledge that further experimental evaluations will be needed to establish the practical advantages of your method in interesting applications.